# Accelerated Stochastic Matrix Inversion: General Theory and Speeding up BFGS Rules for Faster Second-Order Optimization

**Robert M. Gower**
Télécom ParisTech
Paris, France
robert.gower@telecom-paristech.fr

**Filip Hanzely**
KAUST
Thuwal, Saudi Arabia
filip.hanzely@kaust.edu.sa

**Peter Richtárik**[*]
KAUST
Thuwal, Saudi Arabia
peter.richtarik@kaust.edu.sa

**Sebastian U. Stich**
EPFL
Lausanne, Switzerland
sebastian.stich@epfl.ch

## Abstract

We present the first accelerated randomized algorithm for solving linear systems in Euclidean spaces. One essential problem of this type is the matrix inversion problem. In particular, our algorithm can be specialized to invert positive definite matrices in such a way that all iterates (approximate solutions) generated by the algorithm are positive definite matrices themselves. This opens the way for many applications in the field of optimization and machine learning. As an application of our general theory, we develop the *first accelerated (deterministic and stochastic) quasi-Newton updates*. Our updates lead to provably more aggressive approximations of the inverse Hessian, and lead to speed-ups over classical non-accelerated rules in numerical experiments. Experiments with empirical risk minimization show that our rules can accelerate training of machine learning models.

## 1 Introduction

Consider the optimization problem

$$\min_{w \in \mathbb{R}^n} f(w), \tag{1}$$

and assume $f$ is sufficiently smooth. A new wave of second order stochastic methods are being developed with the aim of solving large scale optimization problems. In particular, many of these new methods are based on stochastic BFGS updates [29, 35, 20, 21, 6, 8, 3]. Here we develop a new stochastic accelerated BFGS update that can form the basis of new stochastic quasi-Newton methods.

Another approach to scaling up second order methods is to use randomized *sketching* to reduce the dimension, and hence the complexity of the Hessian and the updates involving the Hessian [26, 38], or *subsampled* Hessian matrices when the objective function is a sum of many loss functions [5, 2, 1, 37].

The starting point for developing second order methods is arguably Newton's method, which performs the iterative process

$$w_{k+1} = w_k - (\nabla^2 f(w_k))^{-1} \nabla f(w_k), \tag{2}$$

---

[*]University of Edinburgh, Moscow Institute of Physics and Technology

where $\nabla^2 f(w_k)$ and $\nabla f(w_k)$ are the Hessian and gradient of $f$, respectively. However, it is inefficient for solving large scale problems as it requires the computation of the Hessian and then solving a linear system at each iteration. Several methods have been developed to address this issue, based on the idea of approximating the exact update.

*Quasi-Newton* methods, in particular BFGS [4, 10, 11, 30], have been the leading optimization algorithm in various fields since the late 60's until the rise of big data, which brought a need for simpler first order algorithms. It is well known that Nesterov's acceleration [22] is a reliable way to speed up first order methods. However until now, acceleration techniques have been applied exclusively to speeding up gradient updates. In this paper we present an accelerated BFGS algorithm, opening up new applications for acceleration. The acceleration in fact comes from an accelerated algorithm for inverting the Hessian matrix.

To be more specific, recall that quasi-Newton rules aim to maintain an estimate of the inverse Hessian $X_k$, adjusting it every iteration so that the inverse Hessian acts appropriately in a particular direction, while enforcing symmetry:

$$X_k(\nabla f(w_k) - \nabla f(w_{k-1})) = w_k - w_{k-1}, \qquad X_k = X_k^\top. \tag{3}$$

A notable research direction is the development of stochastic quasi-Newton methods [15], where the estimated inverse is equal to the true inverse over a subspace:

$$X_k \nabla^2 f(w_k) S_k = S_k, \qquad X_k = X_k^\top, \tag{4}$$

where $S_k \in \mathbb{R}^{n \times \tau}$ is a randomly generated matrix.

In fact, (4) can be seen as the so called sketch-and-project iteration for inverting $\nabla^2 f(w_k)$. In this paper we first develop the accelerated algorithm for inverting positive definite matrices. As a direct application, our algorithm can be used as a primitive in quasi-Newton methods which results in a novel accelerated (stochastic) quasi-Newton method of the type (4). In addition, our acceleration technique can also be incorporated in the classical (non stochastic) BFGS method. This results in the accelerated BFGS method. Whereas the matrix inversion contribution is accompanied by strong theoretical justifications, this does not apply to the latter. Rather, we verify the effectiveness of this new accelerated BFGS method through numerical experiments.

## 1.1 Sketch-and-project for linear systems

Our accelerated algorithm can be applied to more general tasks than only inverting matrices. In its most general form, it can be seen as an accelerated version of a *sketch-and-project* method in Euclidean spaces which we present now. Consider a linear system $Ax = b$ such that $b \in \mathbf{Range}\,(A)$. One step of the sketch-and-project algorithm reads as:

$$x_{k+1} = \operatorname{argmin}_x \|x_k - x\|_B^2 \quad \text{subject to} \quad S_k^\top A x = S_k^\top b, \tag{5}$$

where $\|x\|_B^2 = \langle Bx, x \rangle$ for some $B \succ 0$ and $S_k$ is a random sketching matrix sampled i.i.d at each iteration from a fixed distribution.

Randomized Kaczmarz [16, 33] was the first algorithm of this type. In [13], this sketch-and-project algorithm was analyzed in its full generality. Note that the dual problem of (5) takes the form of a quadratic minimization problem [14], and randomized methods such as coordinate descent [23, 36], random pursuit [31, 32] or stochastic dual ascent [14] can thus also be captured as special instances of this method. Richtárik and Takáč [28] adopt a new point of view through a theory of stochastic reformulations of linear systems. In addition, they consider the addition of a relaxation parameter, as well as mini-batch and accelerated variants. Acceleration was only achieved for the expected iterates, and not in the L2 sense as we do here. We refer to Richtárik and Takáč [28] for interpretation of sketch-and-project as stochastic gradient descent, stochastic Newton, stochastic proximal point method, and stochastic fixed point method.

Gower [15] observed that the procedure (5) can also be applied to find the inverse of a matrix. Assume the optimization variable itself is a matrix, $x = X$, $b = I$, the identity matrix, then sketch-and-project converges (under mild assumptions) to a solution of $AX = I$. Even the symmetry constraint $X = X^\top$ can be incorporated into the sketch-and-project framework since it is a linear constraint.

There has been recent development in speeding up the sketch-and-project method using the idea of Nesterov's acceleration [22]. In [18] an accelerated Kaczmarz algorithm was presented for special

sketches of rank one. Arbitrary sketches of rank one where considered in [31], block sketches in [24] and recently, Tu and coathors [34] developed acceleration for special sketching matrices, assuming the matrix $A$ is square. This assumption, along with any assumptions on $A$, was later dropped in [27]. Another notable way to accelerate the sketch-and-project algorithm is by using momentum or stochastic momentum [19].

We build on recent work of Richtárik and Takáč [27] and further extend their analysis by studying accelerated sketch-and-project in general Euclidean spaces. This allows us to deduce the result for matrix inversion as a special case. However, there is one additional caveat that has to be considered for the intended application in quasi-Newton methods: ideally, all iterates of the algorithm should be symmetric positive definite matrices. This is not the case in general, but we address this problem by constructing special sketch operators that preserve symmetry and positive definiteness.

## 2   Contributions

We now present our main contributions.

**Accelerated Sketch and Project in Euclidean Spaces.** We generalize the analysis of an accelerated version of the sketch-and-project algorithm [27] to linear operator systems in Euclidean spaces. We provide a self-contained convergence analysis, recovering the original results in a more general setting.

**Faster Algorithms for Matrix Inversion.** We develop an accelerated algorithm for inverting positive definite matrices. This algorithm can be seen as a special case of the accelerated sketch-and-project in Euclidean space, thus its convergence follows from the main theorem. However, we also provide a different formulation of the proof that is specialized to this setting. Similarly to [34], the performance of the algorithm depends on two parameters $\mu$ and $\nu$ that capture spectral properties of the input matrix and the sketches that are used. Whilst for the non-accelerated sketch-and-project algorithm for matrix inversion [15] the knowledge of these parameters is not necessary, they need to be given as input to the accelerated scheme. When employed with the correct choice of parameters, the accelerated algorithm is always faster than the non-accelerated one. We also provide a theoretical rate for sub-optimal parameters $\mu, \nu$, and we perform numerical experiments to argue the choice of $\mu, \nu$ in practice.

**Randomized Accelerated Quasi-Newton.** The proposed iterative algorithm for matrix inversion is designed in such a way that each iterate is a symmetric matrix. This means, we can use the generated approximate solutions as estimators for the inverse Hessian in quasi-Newton methods, which is a direct extension of stochastic quasi-Newton methods. To the best of our knowledge, this yields the first accelerated (stochastic) quasi-Newton method.

**Accelerated Quasi-Newton.** In the standard BFGS method the updates to the Hessian estimate are not chosen randomly, but deterministically. Based on the intuition gained from the accelerated random method, we propose an accelerated scheme for BFGS. The main idea is that we replace the random sketching of the Hessian with a deterministic update. The theoretical convergence rates do not transfer to this scheme, but we demonstrate by numerical experiments that it is possible to choose a parameter combination which yields a slightly faster convergence. We believe that the novel idea of accelerating BFGS update is extremely valuable, as until now, acceleration techniques were only considered to improve gradient updates.

### 2.1   Outline

Our accelerated sketch-and-project algorithm for solving linear systems in Euclidean spaces is developed and analyzed in Section 3, and is used later in Section 4 to analyze an accelerated sketch-and-project algorithm for matrix inversion. The accelerated sketch-and-project algorithm for matrix inversion is then used to accelerate the BFGS update, which in turn leads to the development of an accelerated BFGS optimization method. Lastly in Section 5, we perform numerical experiments to gain different insights into the newly developed methods. Proofs of all results and additional insights can be found in the appendix.

# 3 Accelerated Stochastic Algorithm for Matrix Inversion

In this section we propose an accelerated randomized algorithm to solve linear systems in Euclidean spaces. This is a very general problem class which comprises the matrix inversion problem as well. Thus, we will use the result of this section later to analyze our newly proposed matrix inversion algorithm, which we then use to estimate the inverse of the Hessian within a quasi-Newton method.[2]

Let $\mathcal{X}$ and $\mathcal{Y}$ be finite dimensional Euclidean spaces and let $\mathcal{A} : \mathcal{X} \mapsto \mathcal{Y}$ be a linear operator. Let $L(\mathcal{X}, \mathcal{Y})$ denote the space of linear operators that map from $\mathcal{X}$ to $\mathcal{Y}$. Consider the linear system

$$\mathcal{A}x = b, \tag{6}$$

where $x \in \mathcal{X}$ and $b \in \mathbf{Range}\,(\mathcal{A})$. Consequently there exists a solution to the equation (6). In particular, we aim to find the solution closest to a given initial point $x_0 \in \mathcal{X}$:

$$x^* \stackrel{\text{def}}{=} \arg\min_{x \in \mathcal{X}} \tfrac{1}{2}\|x - x_0\|^2 \quad \text{subject to} \quad \mathcal{A}x = b. \tag{7}$$

Using the pseudoinverse and Lemma 22 item *vi*, the solution to (7) is given by

$$x^* = x_0 - \mathcal{A}^\dagger(\mathcal{A}x_0 - b) \in x_0 + \mathbf{Range}\,(\mathcal{A}^*), \tag{8}$$

where $A^\dagger$ and $A^*$ denote the pseudoinverse and the adjoint of $A$, respectively.

## 3.1 The algorithm

Let $\mathcal{Z}$ be a Euclidean space and consider a random linear operator $\mathcal{S}_k \in L(\mathcal{Y}, \mathcal{Z})$ chosen from some distribution $\mathcal{D}$ over $L(\mathcal{Y}, \mathcal{Z})$ at iteration $k$. Our method is given in Algorithm 1, where $Z_k \in L(\mathcal{X})$ is a random linear operator given by the following compositions

$$Z_k = Z(\mathcal{S}_k) \stackrel{\text{def}}{=} \mathcal{A}^*\mathcal{S}_k^*(\mathcal{S}_k\mathcal{A}\mathcal{A}^*\mathcal{S}_k^*)^\dagger \mathcal{S}_k\mathcal{A}. \tag{9}$$

The updates of variables $g_k$ and $x_{k+1}$ on lines 8 and 9, respectively, correspond to what is known as the *sketch-and-project* update:

$$x_{k+1} = \arg\min_{x \in \mathcal{X}} \tfrac{1}{2}\|x - y_k\|^2 \quad \text{subject to} \quad \mathcal{S}_k\mathcal{A}x = \mathcal{S}_k b, \tag{10}$$

which can also be written as the following operation

$$x_{k+1} - x_* = (I - Z_k)(y_k - x_*). \tag{11}$$

This follows from the fact that $b \in \mathbf{Range}\,(\mathcal{A})$, together with item i of Lemma 22. Furthermore, note that the adjoint $\mathcal{A}^*$ and the pseudoinverse in Algorithm 1 are taken with respect to the norm in (7).

---

**Algorithm 1** Accelerated Sketch-and-Project for solving (10) [27]

---

1: **Parameters:** $\mu, \nu > 0$, $\mathcal{D}$ = distribution over random linear operators.

2: Choose $x_0 \in \mathcal{X}$ and set $v_0 = x_0$, $\beta = 1 - \sqrt{\frac{\mu}{\nu}}$, $\gamma = \sqrt{\frac{1}{\mu\nu}}$, $\alpha = \frac{1}{1+\gamma\nu}$.

3: **for** $k = 0, 1, \ldots$ **do**

4:      $y_k = \alpha v_k + (1 - \alpha)x_k$

5:      Sample an independent copy $S_k \sim \mathcal{D}$

6:      $g_k = \mathcal{A}^*\mathcal{S}_k^*(\mathcal{S}_k\mathcal{A}\mathcal{A}^*\mathcal{S}_k^*)^\dagger \mathcal{S}_k(\mathcal{A}y_k - b) = Z_k(y_k - x_*)$

7:      $x_{k+1} = y_k - g_k$

8:      $v_{k+1} = \beta v_k + (1 - \beta)y_k - \gamma g_k$

9: **end for**

---

Algorithm 1 was first proposed and analyzed by Richtárik and Takáč [27] for the special case when $\mathcal{X} = \mathbb{R}^n$ and $\mathcal{Y} = \mathbb{R}^m$. Our contribution here is in extending the algorithm and analysis to the more abstract setting of Euclidean spaces. In addition, we provide some further extensions of this method in Sections D and E, allowing for a non-unit stepsize and variable $\alpha$, respectively.

## 3.2 Key assumptions and quantities

Denote $Z = Z(\mathcal{S})$ for $\mathcal{S} \sim \mathcal{D}$. Assume that the *exactness property* holds

$$\mathbf{Null}\,(\mathcal{A}) = \mathbf{Null}\,(\mathbf{E}\,[Z])\,; \tag{12}$$

this is also equivalent to $\mathbf{Range}\,(\mathcal{A}^*) = \mathbf{Range}\,(\mathbf{E}\,[Z])$. The exactness assumption is of key importance in the sketch-and-project framework, and indeed it is not very strong. For example, it holds for the matrix inversion problem with every sketching strategy we consider. We further assume that $\mathcal{A} \neq 0$ and $\mathbf{E}\,[Z]$ is finite. First we collect a few observation on the $Z$ operator

**Lemma 1.** *The $Z$ operator* (9) *is a self-adjoint positive projection. Consequently* $\mathbf{E}\,[Z]$ *is a self-adjoint positive operator.*

The two parameters that govern the acceleration are

$$\mu \stackrel{\text{def}}{=} \inf_{x \in \mathbf{Range}(\mathcal{A}^*)} \frac{\langle \mathbf{E}[Z]x,x \rangle}{\langle x,x \rangle}, \qquad \nu \stackrel{\text{def}}{=} \sup_{x \in \mathbf{Range}(\mathcal{A}^*)} \frac{\langle \mathbf{E}[Z\mathbf{E}[Z]^\dagger Z]x,x \rangle}{\langle \mathbf{E}[Z]x,x \rangle}. \tag{13}$$

The supremum in the definition of $\nu$ is well defined due to the exactness assumption together with $\mathcal{A} \neq 0$.

**Lemma 2.** *We have*

$$1 \quad \leq \quad \nu \quad \leq \quad \frac{1}{\mu} \quad = \quad \|\mathbf{E}\,[Z]^\dagger\|. \tag{14}$$

*Moreover, if* $\mathbf{Range}\,(\mathcal{A}^*) = \mathcal{X}$, *we have*

$$\frac{\mathbf{Rank}(\mathcal{A}^*)}{\mathbf{E}[\mathbf{Rank}(Z)]} \leq \nu. \tag{15}$$

## 3.3 Convergence and change of the norm

For a positive self-adjoint $G \in L(\mathcal{X})$ and $x \in \mathcal{X}$ let $\|x\|_G \stackrel{\text{def}}{=} \sqrt{\langle x,x \rangle_G} \stackrel{\text{def}}{=} \sqrt{\langle Gx,x \rangle}$. We now informally state the convergence rate of Algorithm 1. Theorem 3 generalizes the main theorem from [27] to linear systems in Euclidean spaces.

**Theorem 3.** *Let* $x_k, v_k$ *be the random iterates of Algorithm 1. Then*

$$\mathbf{E}\left[\|v_k - x_*\|_{\mathbf{E}[Z]^\dagger}^2 + \tfrac{1}{\mu}\|x_k - x_*\|^2\right] \leq \left(1 - \sqrt{\tfrac{\mu}{\nu}}\right)^k \mathbf{E}\left[\|v_0 - x_*\|_{\mathbf{E}[Z]^\dagger}^2 + \tfrac{1}{\mu}\|x_0 - x_*\|^2\right].$$

This theorem shows the accelerated Sketch-and-Project algorithm converges linearly with a rate of $\left(1 - \sqrt{\tfrac{\mu}{\nu}}\right)$, which translates to a total of $O(\sqrt{\nu/\mu}\log(1/\epsilon))$ iterations to bring the given error in Theorem 3 below $\epsilon > 0$. This is in contrast with the non-accelerated Sketch-and-Project algorithm which requires $O((1/\mu)\log(1/\epsilon))$ iterations, as shown in [13] for solving linear systems. From (14), we have the bounds $1/\sqrt{\mu} \leq \sqrt{\nu/\mu} \leq 1/\mu$. On one extreme, this inequality shows that the iteration complexity of the accelerated algorithm is at least as good as its non-accelerated counterpart. On the other extreme, the accelerated algorithm might require as little as the square root of the number of iterations of its non-accelerated counterpart. Since the cost of a single iteration of the accelerated algorithm is of the same order as the non-accelerated algorithm, this theorem shows that acceleration can offer a significant speed-up, which is verified numerically in Section 5. It is also possible to get the convergence rate of accelerated sketch-and-project where projections are taken with respect to a different weighted norm. For technical details, see Section B.4 of the Appendix.

## 3.4 Coordinate sketches with convenient probabilities

Let us consider a simple example in the setting for Algorithm 1 where we can understand parameters $\mu, \nu$. In particular, consider a linear system $Ax = b$ in $\mathbb{R}^n$ where $A$ is symmetric positive definite.

**Corollary 4.** *Choose* $B = A$ *and* $S = e_i$ *with probability proportional to* $A_{i,i}$. *Then*

$$\mu = \frac{\lambda_{\min}(A)}{\mathbf{Tr}(A)} =: \mu^P \quad \text{and} \quad \nu = \frac{\mathbf{Tr}(A)}{\min_i A_{i,i}} =: \nu^P \tag{16}$$

*and therefore the convergence rate given in Theorem 3 for the accelerated algorithm is*

$$\left(1 - \sqrt{\tfrac{\mu}{\nu}}\right)^k \quad = \quad \left(1 - \frac{\sqrt{\lambda_{\min}(A)\min_i A_{i,i}}}{\mathbf{Tr}(A)}\right)^k. \tag{17}$$

Rate (17) of our accelerated method is to be contrasted with the rate of the non-accelerated method: $(1 - \mu)^k = (1 - \lambda_{\min}(A)/\mathbf{Tr}\,(A)))^k$. Clearly, we gain from acceleration if the smallest diagonal element of $A$ is significantly larger than the smallest eigenvalue.

In fact, parameters $\mu^P, \nu^P$ above are the correct choice for the matrix inversion algorithm, when symmetry is not enforced, as we shall see later. Unfortunately, we are not able to estimate the parameters while enforcing symmetry for different sketching strategies. We dedicate a section in numerical experiments to test, if the parameter selection (16) performs well under enforced symmetry and different sketching strategies, and also how one might safely choose $\mu, \nu$ in practice.

# 4 Accelerated Stochastic BFGS Update

The update of the inverse Hessian used in quasi-Newton methods (e.g., in BFGS) can be seen as a sketch-and-project update applied to the linear system $AX = I$, while $X = X^\top$ is enforced, and where $A$ denotes and approximation of the Hessian. In this section, we present an accelerated version of these updates. We provide two different proofs: one based on Theorem 3 and one based on vectorization. By mimicking the updates of the accelerated stochastic BFGS method for inverting matrices, we determine a heuristic for accelerating the classic deterministic BFGS update. We then incorporate this acceleration into the classic BFGS optimization method and show that the resulting algorithm can offer a speed-up of the standard BFGS algorithm.

## 4.1 Accelerated matrix inversion

Consider the symmetric positive definite matrix $A \in \mathbb{R}^{n \times n}$ and the following projection problem

$$A^{-1} = \arg\min_X \|X\|_{F(A)}^2 \quad \text{subject to} \quad AX = I, \quad X = X^\top, \tag{18}$$

where $\|X\|_{F(A)} \stackrel{\text{def}}{=} \mathbf{Tr}\,(AX^\top AX) = \|A^{1/2}XA^{1/2}\|_F^2$. This projection problem can be cast as an instantiation of the general projection problem (7). Indeed, we need only note that the constraint in (18) is linear and equivalent to $\mathcal{A}(X) \stackrel{\text{def}}{=} \left(\begin{smallmatrix} AX \\ X - X^\top \end{smallmatrix}\right) = \left(\begin{smallmatrix} I \\ 0 \end{smallmatrix}\right)$. The matrix inversion problem can be efficiently solved using sketch-and-project with a symmetric sketch [15]. The symmetric sketch is given by $\mathcal{S}_k \mathcal{A}(X) = \left(\begin{smallmatrix} S_k^\top AX \\ X - X^\top \end{smallmatrix}\right)$, where $S_k \in \mathbb{R}^{n \times \tau}$ is a random matrix drawn from a distribution $\mathcal{D}$ and $\tau \in \mathbb{N}$. The resulting sketch-and-project method is as follows

$$X_{k+1} = \arg\min_X \|X - X_k\|_{F(A)}^2 \quad \text{subject to} \quad S_k^\top AX = S_k^\top, \quad X = X^\top, \tag{19}$$

the closed form solution of which is

$$X_{k+1} = S_k(S_k^\top AS_k)^{-1}S_k^\top + \left(I - S_k(S_k^\top AS_k)^{-1}S_k^\top A\right) X_k \left(I - AS_k(S_k^\top AS_k)^{-1}S_k^\top\right). \tag{20}$$

By observing that (20) is the sketch-and-project algorithm applied to a linear operator equation, we have constructed an accelerated version in Algorithm 2. We can also apply Theorem 3 to prove that Algorithm 2 is indeed accelerated.

**Theorem 5.** *Let* $L^k \stackrel{\text{def}}{=} \|V_k - A^{-1}\|_M^2 + \frac{1}{\mu}\|X_k - A^{-1}\|_{F(A)}^2$. *The iterates of Algorithm 2 satisfy*

$$\mathbf{E}\,[L_{k+1}] \leq \left(1 - \sqrt{\tfrac{\mu}{\nu}}\right) \mathbf{E}\,[L_k], \tag{21}$$

*where* $\|X\|_M^2 = \mathbf{Tr}\left(A^{1/2}X^\top A^{1/2}\mathbf{E}\,[Z]^\dagger A^{1/2}XA^{1/2}\right)$. *Furthermore,*

$$\mu \stackrel{\text{def}}{=} \inf_{X \in \mathbb{R}^{n \times n}} \frac{\langle \mathbf{E}[Z]X, X\rangle}{\langle X, X\rangle} = \lambda_{\min}(\mathbf{E}\,[Z]), \qquad \nu \stackrel{\text{def}}{=} \sup_{X \in \mathbb{R}^{n \times n}} \frac{\langle \mathbf{E}\left[Z\mathbf{E}[Z]^\dagger Z\right]X, X\rangle}{\langle \mathbf{E}[Z]X, X\rangle}, \tag{22}$$

*where*

$$\mathbf{Z} \stackrel{\text{def}}{=} I \otimes I - (I - P) \otimes (I - P), \qquad P \stackrel{\text{def}}{=} A^{1/2}S(S^\top AS)^{-1}S^\top A^{1/2}, \tag{23}$$

*and* $Z : X \in \mathbb{R}^{n \times n} \to \mathbb{R}^{n \times n}$ *is given by* $Z(X) = X - (I - P) X (I - P) = XP + PX(I - P)$. *Moreover,* $2\lambda_{\min}(\mathbf{E}\,[P]) \geq \lambda_{\min}(\mathbf{E}\,[Z]) \geq \lambda_{\min}(\mathbf{E}\,[P])$.

Notice that preserving symmetry yields $\mu = \lambda_{\min}(\mathbf{E}\,[Z])$, which can be up to twice as large as $\lambda_{\min}(\mathbf{E}\,[P])$, which is the value of the $\mu$ parameter of the method without preserving symmetry. This improved rate is new, and was not present in the algorithm's debut publication [15]. In terms of parameter estimation, once symmetry is not preserved, we fall back onto the setting from Section 3.4. Unfortunately, we were not able to quantify the effect of enforcing symmetry on the parameter $\nu$.

---

**Algorithm 2** Accelerated BFGS matrix inversion (solving (18))

---

1: **Parameters:** $\mu, \nu > 0$, $\mathcal{D}$ = distribution over random linear operators.
2: Choose $X_0 \in \mathcal{X}$ and set $V_0 = X_0$, $\beta = 1 - \sqrt{\frac{\mu}{\nu}}$, $\gamma = \sqrt{\frac{1}{\mu\nu}}$, $\alpha = \frac{1}{1+\gamma\nu}$
3: **for** $k = 0, 1, \ldots$ **do**
4: $\quad Y_k = \alpha V_k + (1 - \alpha)X_k$
5: $\quad$ Sample an independent copy $S \sim \mathcal{D}$
6: $\quad X_{k+1} = Y_k + (Y_k A - I)S(S^\top A S)^{-1}S^\top - S(S^\top A S)^{-1}S^\top A Y_k$
7: $\quad\quad + S(S^\top A S)^{-1}S^\top A Y_k A S(S^\top A S)^{-1}S^\top$
8: $\quad V_{k+1} = \beta V_k + (1 - \beta)Y_k - \gamma(Y_k - X_{k+1})$
9: **end for**

---

## 4.2 Vectorizing—a different insight

In the previous section we argued that Theorem 5 follows from the more general convergence result established in Theorem 3 for Euclidean spaces. We now show an alternative way to prove Theorem 5. Define $\mathbf{Vec} : \mathbb{R}^{n \times n} \to \mathbb{R}^{n^2}$ to be a vectorization operator of column-wise stacking and denote $x \stackrel{\text{def}}{=} \mathbf{Vec}(X)$. It can be shown that the sketch-and-project operation for matrix inversion (4.2) is equivalent to

$$x_{k+1} = \arg\min_x \|x - x_k\|^2_{A \otimes A} \quad \text{subject to} \quad (I \otimes S_k^\top)(I \otimes A)x = (I \otimes S_k^\top)\mathbf{Vec}(I), \ Cx = 0,$$

where $C$ is defined so that $Cx = 0$ if and only if $X = X^\top$. The above is a sketch-and-project update for a linear system in $\mathbb{R}^{n^2}$, which allows to obtain an alternative proof of Theorem 5, without using our results from Euclidean spaces. The details are provided in Section H.2 of the Appendix.

## 4.3 Accelerated BFGS as an optimization algorithm

As a tweak in the stochastic BFGS allows for a faster estimation of Hessian inverse and therefore more accurate steps of the method, one might wonder if a equivalent tweak might speed up the standard, deterministic BFGS algorithm for solving (1). The mentioned tweaked version of standard BFGS is proposed as Algorithm 3. We do not state a convergence theorem for this algorithm—due to the deterministic updates the analysis is currently elusive—nor propose to use it as a default solver, but we rather introduce it as a novel idea for accelerating optimization algorithms. We leave theoretical analysis for the future work. For now, we perform several numerical experiments, in order to understand the potential and limitations of this new method.

---

**Algorithm 3** BFGS method with accelerated BFGS update for solving (1)

---

1: **Parameters:** $\mu, \nu > 0$, stepsize $\eta$.
2: Choose $X_0 \in \mathcal{X}$, $w_0$ and set $V_0 = X_0$, $\beta = 1 - \sqrt{\frac{\mu}{\nu}}$, $\gamma = \sqrt{\frac{1}{\mu\nu}}$, $\alpha = \frac{1}{1+\gamma\nu}$.
3: **for** $k = 0, 1, \ldots$ **do**
4: $\quad w_{k+1} = w_k - \eta X_k \nabla f(w_k)$
5: $\quad s_k = w_{k+1} - w_k, \quad \zeta_k = \nabla f(w_{k+1}) - \nabla f(w_k)$
6: $\quad Y_k = \alpha V_k + (1 - \alpha)X_k$
7: $\quad X_{k+1} = \frac{\delta_k \delta_k^\top}{\delta_k^\top \zeta_k} + \left(I - \frac{\delta_k \zeta_k^\top}{\delta_k^\top \zeta_k}\right) Y_k \left(I - \frac{\zeta_k \delta_k^\top}{\delta_k^\top \zeta_k}\right)$
8: $\quad V_{k+1} = \beta V_k + (1 - \beta)Y_k - \gamma(Y_k - X_{k+1})$
9: **end for**

---

To better understand Algorithm 3, recall that the BFGS updates an estimate of the inverse Hessian via

$$X_{k+1} = \text{argmin}_X \|X - X_k\|^2_{F(A)} \quad \text{subject to} \quad X\zeta_k = \delta_k, \ X = X^\top, \tag{24}$$

where $\delta_k = w_{k+1} - w_k$ and $\zeta_k = \nabla f(w_{k+1}) - \nabla f(w_k)$. The above has the following closed form solution $X_{k+1} = \frac{\delta_k \delta_k^\top}{\delta_k^\top \zeta_k} + \left(I - \frac{\delta_k \zeta_k^\top}{\delta_k^\top \zeta_k}\right) X_k \left(I - \frac{\zeta_k \delta_k^\top}{\delta_k^\top \zeta_k}\right)$. This update appears on line 7 of Algorithm 3 with the difference being that it is applied to a matrix $Y_k$.

# 5 Numerical Experiments

We perform extensive numerical experiments to bring additional insight to both the performance of and to parameter selection for Algorithms 2 and 3. More numerical experiments can be found in Section A of the appendix. We first test our accelerated matrix inversion algorithm, and subsequently perform experiments related to Section 4.3.

## 5.1 Accelerated Matrix Inversion

We consider the problem of inverting a symmetric positive matrix $A$. We focus on a few particular choices of matrices $A$ (specified when describing each experiment), that differ in their eigenvalue spectra. Three different sketching strategies are studied: Coordinate sketches with convenient probabilities ($S = e_i$ with probability proportional to $A_{i,i}$), coordinate sketches with uniform probabilities ($S = e_i$ with probability $\frac{1}{n}$) and Gaussian sketches ($S \sim \mathcal{N}(0, I)$). As matrices to be inverted, we use both artificially generated matrices with the access to the spectrum and also Hessians of ridge regression problems from LIBSVM.

We have shown earlier that $\mu, \nu$ can be estimated as per (16) for coordinate sketches with convenient probabilities without enforcing symmetry. We use the mentioned parameters for the other sketching strategies while enforcing the symmetry. Since in practice one might not have an access to the exact parameters $\mu, \nu$ for given sketching strategy, we test sensitivity of the algorithm to parameter choice . We also test test for $\nu$ chosen by (16), $\mu = \frac{1}{100\nu}$ and $\mu = \frac{1}{10000\nu}$.

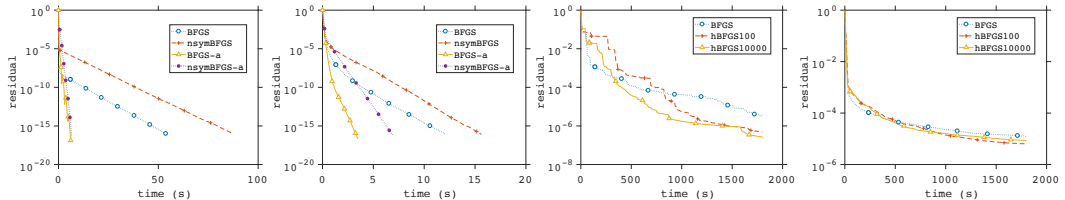

Figure 1: From left to right: (i) Eigenvalues of $A \in \mathbb{R}^{100 \times 100}$ are $1, 10^3, 10^3, \ldots, 10^3$ and coordinate sketches with convenient probabilities are used. (ii) Eigenvalues of $A \in \mathbb{R}^{100 \times 100}$ are $1, 2, \ldots, n$ and Gaussian sketches are used. Label "nsym" indicates non-enforcing symmetry and "-a" indicates acceleration. (iii) Epsilon dataset ($n = 2000$), coordinate sketches with uniform probabilities. (iv) SVHN dataset ($n = 3072$), coordinate sketches with convenient probabilities. Label "h" indicates that $\lambda_{\min}$ was not precomputed, but $\mu$ was chosen as described in the text.

For more plots, see Section A in the appendix as here we provide only a tiny fraction of all plots. The experiments suggest that once the parameters $\mu, \nu$ are estimated exactly, we get a speedup comparing to the nonaccelerated method; and the amount of speedup depends on the structure of $A$ and the sketching strategy. We observe from Figure 1 that we gain a great speedup for ill conditioned problems once the eigenvalues are concentrated around the largest eigenvalue. We also observe from Figure 1 that enforcing symmetry combines well with $\mu, \nu$ computed by (16), which does not consider the symmetry. On top of that, choice of $\mu, \nu$ per (16) seems to be robust to different sketching strategies, and in worst case performs as fast as the nonaccelerated algorithm.

## 5.2 BFGS Optimization Method

We test Algorithm 3 on several logistic regression problems using data from LIBSVM [7]. In all our tests we centered and normalized the data, included a bias term (a linear intercept), and choose the regularization parameter as $\lambda = 1/m$, where $m$ is the number of data points. To keep things as simple as possible, we also used a fixed stepsize which was determined using grid search. Since our theory regarding the choice for the parameters $\mu$ and $\nu$ does not apply in this setting, we simply probed the space of parameters manually and reported the best found result, see Figure 2. In the legend we use BFGS-a-$\mu$-$\nu$ to denote the accelerated BFGS method (Alg 3) with parameters $\mu$ and $\nu$.

On all four datasets, our method outperforms the classic BFGS method, indicating that replacing classic BFGS update rules for learning the inverse Hessian by our new accelerated rules can be beneficial in practice. In A.4 in the appendix we also show the time plots for solving the problems in Figure 2, and show that the accelerated BFGS method also converges faster in time.

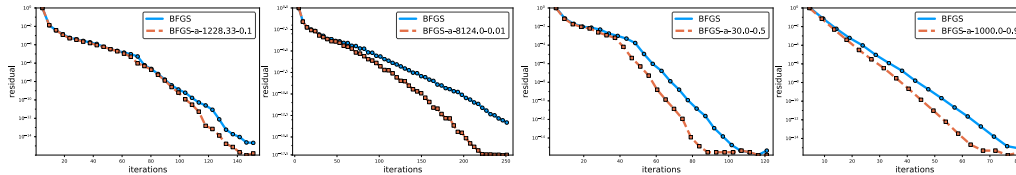

Figure 2: Algorithm 3 (BFGS with accelerated matrix inversion quasi-Newton update) vs standard BFGS. From left to right: `phishing`, `mushrooms`, `australian` and `splice` dataset.

## 6 Conclusions and Extensions

We developed an accelerated sketch-and-project method for solving linear systems in Euclidean spaces. The method was applied to invert positive definite matrices, while keeping their symmetric structure for all iterates. Our accelerated matrix inversion algorithm was then incorporated into an optimization framework to develop both accelerated stochastic and accelerated deterministic BFGS, which to the best of our knowledge, are *the first accelerated quasi-Newton updates.*

We show that under a careful choice of the parameters of the method—depending on the problem structure and conditioning—acceleration might result into significant speedups both for the matrix inversion problem and for the stochastic BFGS algorithm. We confirm experimentally that our accelerated methods can lead to speed-ups when compared to the classical BFGS algorithm.

As a future line of research it might be interesting to study the accelerated BFGS algorithm (either deterministic or stochastic) further, and provide a convergence analysis on a suitable class of functions. Another interesting area of research might be to combine accelerated BFGS with limited memory [17] or engineer the method so that it can efficiently compete with first order algorithms for some empirical risk minimization problems, such as, for example [12].

As we show in this work, *Nesterov's acceleration can be applied to quasi-Newton updates.* We believe this is a surprising fact, as quasi-Newton updates have not been understood as optimization algorithms, which prevented the idea of applying acceleration in this context.

Since since second-order methods are becoming more and more ubiquitous in machine learning and data science, we hope that our work will motivate further advances at the frontiers of big data optimization.

## Footnotes

[2]Quasi-Newton methods do not compute an exact matrix inverse, rather, they only compute an incremental update. Thus, it suffices to apply *one step* of our proposed scheme per iteration. This will be detailed in Section 4.

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
