[Supplementary Material]

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

# A Further Experiments with Accelerated quasi-Newton Updates

In this section, we test the the empirical rate of convergence of Algorithm 2, the accelerated BFGS update for inverting positive definite matrices. Only vector sketches are considered, as the standard quasi-Newton methods also update the inverse Hessian only according to the action in one direction. We compare the speed of the accelerated method with precomputed estimates of the parameters $\mu, \nu$ to the nonaccelerated method. The precomputed estimates of $\mu^P, \nu^P$ are set as per (16):

$$\mu^P = \frac{\lambda_{\min}(A)}{\mathbf{Tr}\,(A)}, \qquad \nu^P = \frac{\mathbf{Tr}\,(A)}{\min_i (A_{i,i})},$$

which is the optimal choice for coordinate sketches with convenient probabilities without enforcing symmetry. In practice we might not have an access to $\lambda_{\min}(A)$, thus we cannot compute $\mu^P$ exactly. Therefore we also test sensitivity of the algorithm to the choice of parameters, and we run some experiments where we only guess parameter $\mu^P$.

Lastly, the tests are performed on both artificial examples and LIBSVM [7] data. We shall also explain the legend of plots: "a" indicates acceleration, "nsym" indicates the algorithm without enforcing symmetry and "h" indicates the setting when $\nu^P$ is not known, and a naive heuristic choice is casted.

## A.1 Simple and well understood artificial example

Let us consider inverting the matrix $A = \alpha I + \beta \mathbf{1}\mathbf{1}^\top$ for $\alpha > 0$ and $\beta \geq -\frac{\alpha}{n}$ so as in this case we have control over both $\mu$ and $\nu$. This artificial example was considered in [34] for solving linear systems. In particular, we show that for coordinate sketches with convenient probabilities (which is indeed the same as uniform probabilities in this example), we have

$$\mu^P \overset{\text{def}}{=} \lambda_{\min}(\mathbf{E}\,[P]) = \frac{\min\,(\alpha, \alpha + n\beta)}{n(\alpha + \beta)},$$

$$\nu^P \overset{\text{def}}{=} \lambda_{\max}\left(\mathbf{E}\left[\mathbf{E}\,[P]^{-\frac{1}{2}}\,P\mathbf{E}\,[P]^{-1}\,P\mathbf{E}\,[P]^{-\frac{1}{2}}\right]\right) = n.$$

Due to the fact that we do not have a theoretical justification of $\mu, \nu$ for $n > 2$ when enforcing symmetry, we set $\mu = \mu^P$ and $\nu = \nu^P$ for Gaussian sketches as well.

Figure 3: Parameter choice: $\alpha = 1 + 10^{-1}, \beta = -n^{-1}, n = 100$. From left to right we have: Coordinate sketch with uniform (convenient) probabilities and Gaussian sketch respectively.

Figure 4: Parameter choice: $\alpha = 1 + 10^{-3}, \beta = -n^{-1}, n = 100$. From left to right we have: Coordinate sketch with uniform (convenient) probabilities and Gaussian sketch respectively.

Figure 5: Parameter choice: $\alpha = 1 + 10^{-5}, \beta = -n^{-1}, n = 100$. From left to right we have: Coordinate sketch with uniform (convenient) probabilities and Gaussian sketch, respectively.

As expected from the theory, as the matrix to be inverted becomes more ill conditioned, the accelerated method performs significantly better compared to the nonaccelerated method for coordinate sketches. In fact, an arbitrary speedup can be obtained by setting $\beta = -n^{-1}$ and $\alpha \to 1$ for the coordinate sketches setup. On the other hand, Gaussian sketches report the slowing of the algorithm, most likely caused by the fact that the theoretical parameters $\mu, \nu$ for Gaussian sketches with enforced symmetry are different to $\mu^P, \nu^P$, which are estimated for coordinate sketches without enforced symmetry. In the case of coordinate sketches with symmetry enforced, we suspect a great speedup even though the parameters $\mu, \nu$ were set to $\mu^P, \nu^P$.

### A.2 Random artificial example

We randomly generate an orthonormal matrix $U$, choose diagonal matrix $D$, and set $A = UDU^\top$. Clearly, diagonal elements of $D$ are eigenvalues of $A$. We set them in the following way:

- Uniform grid. The eigenvalues are set to $1, 2, \ldots, n$.

- One small, the rest larger. The smallest eigenvalue is 1, remaining eigenvalues are all 10 in the first example, all 100 in the second example and all 1000 in the third example in this category.

- One large, the rest small. The largest eigenvalue is $10^4$, the remaining eigenvalues are all 1.

Firstly, consider coordinate sketches with convenient probabilities. Notice that we can easily estimate $\nu^P, \mu^P$ due to the results from Section 3.4 since we have control of $\lambda_{\min}(A)$ and therefore also of $\mu$. Therefore, we set $\mu = \mu^P = \min D_{i,i}$ and $\nu = \nu^P$ for Algorithm 2. Then, we consider coordinate sketches with uniform probabilities and Gaussian sketches. In both cases, we set the parameters $\mu, \nu$ as for coordinate sketches with convenient probabilities.

Figure 6: Eigenvalues set to $1, 2, 3, \ldots n$. From left to right we have: Coordinate sketch with convenient probabilities, coordinate sketch with uniform probabilities and Gaussian sketch respectively.

Figure 7: Eigenvalues set to $1, 10, 10, \ldots 10$. From left to right we have: Coordinate sketch with convenient probabilities, coordinate sketch with uniform probabilities and Gaussian sketch respectively.

Figure 8: Eigenvalues set to $1, 100, 100, \ldots 100$. From left to right we have: Coordinate sketch with convenient probabilities, coordinate sketch with uniform probabilities and Gaussian sketch respectively.

Figure 9: Eigenvalues set to $1, 1000, 1000, \ldots 1000$. From left to right we have: Coordinate sketch with convenient probabilities, coordinate sketch with uniform probabilities and Gaussian sketch respectively.

Figure 10: Eigenvalues set to $10000, 1, 1, \ldots 1$. From left to right we have: Coordinate sketch with convenient probabilities, coordinate sketch with uniform probabilities and Gaussian sketch respectively.

The numerical experiments in this section indicate that one might choose $\mu, \nu$ as per Section 3.4. In other words, one might pretend to be in the setting when symmetry is not enforced and coordinate sketches with convenient probabilities are used. In fact, the practical speedup coming from the acceleration depends very strongly on the structure of matrix $A$. Another message to be delivered is that both preserving symmetry and acceleration yield a better convergence and they combine together well.

We also consider a problem where we pretend to not have access to $\lambda_{\min}(A)$, therefore we cannot choose $\mu = \mu^P$. Instead, we naively choose $\mu = \frac{1}{100\nu}$ and $\mu = \frac{1}{10000\nu}$.

Figure 11: Eigenvalues set to $1, 2, \ldots, n$. From left to right we have: Coordinate sketch with convenient probabilities, coordinate sketch with uniform probabilities and Gaussian sketch respectively.

Figure 12: Eigenvalues set to $1, 10, 10, \ldots 10$. Coordinate sketch with convenient probabilities, coordinate sketch with uniform probabilities and Gaussian sketch respectively.

Figure 13: Eigenvalues set to $1, 100, 100, \ldots 100$. From left to right we have: Coordinate sketch with convenient probabilities, coordinate sketch with uniform probabilities and Gaussian sketch respectively.

Figure 14: Eigenvalues set to $1, 1000, 1000, \ldots 1000$. From left to right we have: Coordinate sketch with convenient probabilities, coordinate sketch with uniform probabilities and Gaussian sketch respectively.

Figure 15: Eigenvalues set to $10000, 1, 1, \ldots 1$. From left to right we have: Coordinate sketch with convenient probabilities, coordinate sketch with uniform probabilities and Gaussian sketch respectively.

Notice that once the acceleration parameters are not set exactly (but they are still reasonable), we observe that the performance of the accelerated algorithm is essentially the same as the performance of the nonaccelerated algorithm. We have observed the similar behavior when setting $\mu = \mu^P$ for Gaussian sketches.

### A.2.1 Sensitivity to the acceleration parameters

Here we investigate the sensitivity of the accelerated BFGS to the parameters $\mu$ and $\nu$. First we compute $\nu^P, \mu^P$ and from this we extract the following exponential grids: $\mu_i = 2^{i-4}\mu$ and $\nu_i = 5^{i-4}\nu$ for $i = 1, 2, \ldots 7$. To gauge the gain is using acceleration with a particular $(\mu, \nu)$ pair, we run the accelerated algorithm for a fixed time then store the error of the final iterate. We then compute average per iteration decrease and divide it by average per iteration decrease of nonaccelerated algorithm. Thus if the resulting difference is less than one, then the accelerated algorithm was faster to nonaccelerated.

In the plots below, $n = 200$ was chosen. We focused on 2 problems described in the previous section—when the eigenvalues are uniformly distributed and when the the largest eigenvalue have multiplicity $n - 1$.

Figure 16: Sensitivity to acceleration parameters. Eigenvalues of $A$ are set to $1, 2 \ldots, n$. From left to right we have: Coordinate sketches with convenient probabilities, coordiante sketches with uniform probabilities and Gaussian sketches. Choice of parameters as per (16) in the middle of plots. Each instance was run for 5 seconds.

Figure 17: Sensitivity to acceleration parameters. Eigenvalues of $A$ are set to $1, 10, 10, \ldots, 10$. From left to right we have: Coordinate sketches with convenient probabilities, coordiante sketches with uniform probabilities and Gaussian sketches. Choice of parameters as per (16) in the middle of plots. Each instance was run for 2 seconds.

Figure 18: Sensitivity to acceleration parameters. Eigenvalues of $A$ are set to $1, 1000, 1000, \ldots, 1000$. From left to right we have: Coordinate sketches with convenient probabilities, coordiante sketches with uniform probabilities and Gaussian sketches. Choice of parameters as per (16) in the middle of plots. Each instance was run for 10 seconds.

The crucial aspect to make the accelerated algorithm to converge is to set $\nu$ large enough. In fact, combination of both small $\nu$ and small $\mu$ leads almost always to non-convergent algorithm. On the other hand, it seems that once $\nu$ is chosen correctly, big enough $\mu$ leads to fast convergence. This indicates how to compute $\mu$ in practice (recall that computing $\nu$ is feasible)—one needs just to choose it small enough (definitely smaller than $\frac{1}{\nu}$).

### A.3 Experiments with LIBSVM

Next we investigate if the accelerated BFGS update improves upon the standard BFGS update when applied to the Hessian $\nabla^2 f(x)$ of ridge regression problems of the form

$$\min_{x \in \mathbb{R}^n} f(x) \overset{\text{def}}{=} \frac{1}{2}\|Ax - b\|_2^2 + \frac{\lambda}{2}\|x\|_2^2, \qquad \nabla^2 f(x) = A^\top A + \lambda I, \tag{25}$$

using data from LIBSVM [7]. Datapoints (rows of $A$) were normalized such that $\|A_{i:}\|^2 = 1$ for all $i$ and the regularization parameter was chosen as $\lambda = \frac{1}{m}$.

First, we run the experiments on smaller problems when parameters $\mu$, $\nu$ are precomputed for coordinate sketches with convenient probabilities (16).

Figure 19: Dataset aloi: $n = 128$. From left to right we have: Coordinate sketch with convenient probabilities, coordinate sketch with uniform probabilities and Gaussian sketch respectively.

Figure 20: Dataset w1a: $n = 300$. From left to right we have: Coordinate sketch with convenient probabilities, coordinate sketch with uniform probabilities and Gaussian sketch respectively.

Figure 21: Dataset w2a: $n = 300$. From left to right we have: Coordinate sketch with convenient probabilities, coordinate sketch with uniform probabilities and Gaussian sketch respectively.

Figure 22: Dataset mushrooms: $n = 112$. From left to right we have: Coordinate sketch with convenient probabilities, coordinate sketch with uniform probabilities and Gaussian sketch respectively.

Figure 23: Dataset protein: $n = 357$. From left to right we have: Coordinate sketch with convenient probabilities, coordinate sketch with uniform probabilities and Gaussian sketch respectively.

Figure 24: Dataset phishing: $n = 68$. From left to right we have: Coordinate sketch with convenient probabilities, coordinate sketch with uniform probabilities and Gaussian sketch respectively.

In the vast majority of examples, the accelerated method performed significantly better than the nonaccelerated method for coordinate sketches (with both convenient and uniform probabilities), however the methods were comparable for Gaussian sketches. We believe that this is due to the fact that choice of parameters as per (16) is close to the optimal parameters for coordinate sketches, and further for Gaussian sketches. However, the experiments on coordinate sketches indicates that for some classes of problems, accelerated algorithms with finely tuned parameters bring a great speedup compared to nonaccelerated ones.

We also consider a problem where we do not compute $\lambda_{\min}(A)$, and therefore we cannot choose $\mu = \mu^P$ in (16). Instead, we choose $\mu = \frac{1}{100\nu}$ and $\mu = \frac{1}{10000\nu}$.

Figure 25: Dataset madelon: $n = 500$. From left to right we have: Coordinate sketch with convenient probabilities, coordinate sketch with uniform probabilities and Gaussian sketch respectively.

Figure 26: Dataset epsilon: $n = 2000$. From left to right we have: Coordinate sketch with convenient probabilities, coordinate sketch with uniform probabilities and Gaussian sketch respectively.

Figure 27: Dataset svhn: $n = 3072$. From left to right we have: Coordinate sketch with convenient probabilities, coordinate sketch with uniform probabilities and Gaussian sketch respectively.

Figure 28: Dataset gisette: $n = 5000$. From left to right we have: Coordinate sketch with convenient probabilities, coordinate sketch with uniform probabilities and Gaussian sketch respectively.

Notice that once the acceleration parameters are not set exactly (but they are still reasonable), we observe that the performance of the accelerated algorithm is essentially the same as the performance of the nonaccelerated algorithm, which is essentially the same conclusion as for artificially generated examples.

### A.4 Additional optimization experiments

In Figure 29 we solve the same problems with the same setup as in 29, but now we plot the time versus the residual (as opposed to iterations versus the residual). Despite the more costly iterations, the accelerated BFGS method can still converge faster than the classic BFGS method.

Figure 29: Algorithm 3 (BFGS with accelerated matrix inversion quasi-Newton update) vs standard BFGS. From left to right: `phishing`, `mushrooms`, `australian` and `splice` dataset.

We also give additional experiments with the same setup to the ones found in Section 5.2. Much like the `phishing` problem in Figure 2, the problems `madelon`, `covtype` and `a9a` in Figures 30, 31 and 32 did not benefit that much from acceleration. Indeed, we found in our experiments that even when choosing extreme values of $\mu$ and $\nu$, the generated inverse Hessian would not significantly deviate from the estimate that one would obtain using the standard BFGS update. Thus on these two problems there is apparently little room for improvement by using acceleration.

Figure 30: `madelon`:   Figure 31: `covtype`   Figure 32: `a9a`

# B   Proofs for Section 3

## B.1   Proof of Lemma 2

First note that $Z$ is a self-adjoint positive operator and thus so is $\mathbf{E}\left[Z\right]$. Consequently.

$$
\begin{aligned}
\mu \quad &\overset{(13)}{=} \quad \inf_{x\in\mathbf{Range}(\mathcal{A}^*)} \frac{\langle \mathbf{E}\left[Z\right]x,x\rangle}{\langle x,x\rangle} \\[6pt]
&\overset{(12)}{=} \quad \inf_{x\in\mathbf{Range}(\mathbf{E}[Z])} \frac{\langle \mathbf{E}\left[Z\right]x,x\rangle}{\langle x,x\rangle} \\[6pt]
&\overset{\text{Lemma 22 item ii}}{=} \quad \inf_{x\in\mathcal{X}} \frac{\langle \mathbf{E}\left[Z\right]\mathbf{E}\left[Z\right]^{\dagger}x, \mathbf{E}\left[Z\right]^{\dagger}x\rangle}{\langle \mathbf{E}\left[Z\right]^{\dagger}x, \mathbf{E}\left[Z\right]^{\dagger}x\rangle} \\[6pt]
&\overset{\text{Lemma 22 item i}}{=} \quad \inf_{x\in\mathcal{X}} \frac{\langle \mathbf{E}\left[Z\right]^{\dagger}x, x\rangle}{\langle \mathbf{E}\left[Z\right]^{\dagger}x, \mathbf{E}\left[Z\right]^{\dagger}x\rangle} \\[6pt]
&\overset{\text{Lemma 18}}{=} \quad \inf_{z\in\mathbf{Range}\left((\mathbf{E}[Z]^{\dagger})^{1/2}\right)} \frac{\langle z,z\rangle}{\langle \mathbf{E}\left[Z\right]^{\dagger}z,z\rangle} \qquad (\text{set } z = (\mathbf{E}\left[Z\right]^{\dagger})^{1/2}x) \\[6pt]
&\overset{(71)}{=} \quad \frac{1}{\|\mathbf{E}\left[Z\right]^{\dagger}\|}. \quad\quad\quad\quad\quad\quad\quad\quad\quad\quad\quad\quad\quad\quad\quad (26)
\end{aligned}
$$

For the bounds (14) we have that

$$
\begin{aligned}
\nu \quad &\overset{(13)}{=} \quad \sup_{x\in\mathbf{Range}(\mathcal{A}^*)} \frac{\mathbf{E}\left[\langle \mathbf{E}\left[Z\right]^{\dagger}Zx, Zx\rangle\right]}{\langle \mathbf{E}\left[Z\right]x,x\rangle} \\[6pt]
&\leq \quad \sup_{x\in\mathbf{Range}(\mathcal{A}^*)} \frac{\|\mathbf{E}\left[Z\right]^{\dagger}\|\mathbf{E}\left[\|Zx\|_2^2\right]}{\langle \mathbf{E}\left[Z\right]x,x\rangle} \\[6pt]
&= \quad \|\mathbf{E}\left[Z\right]^{\dagger}\| \\[6pt]
&\overset{(26)}{\leq} \quad \frac{1}{\mu}.
\end{aligned}
$$

To bound $\nu$ from below we use that $\mathbf{E}\left[Z\right]^{\dagger}$ is self adjoint together with that the map $X \mapsto \langle X\mathbf{E}\left[Z\right]^{\dagger}Xx,x\rangle$ is convex over the space of self-adjoint operators $X \in L(\mathcal{X})$ and for a fixed $x \in \mathcal{X}$. Consequently by Jensen's inequality

$$
\mathbf{E}\left[\langle Z\mathbf{E}\left[Z\right]^{\dagger}Zx,x\rangle\right] \geq \langle \mathbf{E}\left[Z\right]\mathbf{E}\left[Z\right]^{\dagger}\mathbf{E}\left[Z\right]x,x\rangle \overset{\text{Lemma 22 item i}}{=} \langle \mathbf{E}\left[Z\right]x,x\rangle. \quad (27)
$$

Finally

$$\nu \overset{(27)}{\geq} \sup_{x \in \mathbf{Range}(\mathcal{A}^*)} \frac{\langle \mathbf{E}[Z]x, x \rangle}{\langle \mathbf{E}[Z]x, x \rangle} = 1.$$

Lastly, to show (15) we have

$$
\begin{aligned}
\mathbf{Rank}(\mathcal{A}^*) &\overset{(12)}{=} \mathbf{Rank}(\mathbf{E}[Z]) \\
&\overset{\text{Lemma 17+ Lemma 22 (v)}}{=} \mathbf{Tr}\left( \mathbf{E}[Z]\mathbf{E}[Z]^\dagger \right) = \mathbf{E}\left[ \mathbf{Tr}\left( Z\mathbf{E}[Z]^\dagger \right) \right] \\
&= \mathbf{E}\left[ \mathbf{Tr}\left( Z\mathbf{E}[Z]^\dagger Z \right) \right] \\
&\leq \nu \mathbf{E}\left[ \mathbf{Tr}(Z) \right] \overset{\text{Lemma 17}}{=} \nu \mathbf{E}\left[ \mathbf{Rank}(Z) \right],
\end{aligned}
$$

where we used that $\langle \mathbf{E}\left[ Z\mathbf{E}[Z]^\dagger Z \right] u, u \rangle \leq \nu \langle \mathbf{E}[Z]u, u \rangle$ for every $u \in \mathbf{Range}(\mathbf{E}[Z]) = \mathbf{Range}(\mathcal{A}^*) = \mathcal{X}$. $\qquad\square$

**Proof** that $X \mapsto \langle X\mathbf{E}[Z]^\dagger Xx, x \rangle = \|Xx\|^2_{\mathbf{E}[Z]^\dagger}$ is convex: Let $G = \mathbf{E}[Z]^\dagger$ then

$$
\begin{aligned}
\|(\lambda X + (1-\lambda)Y)x\|^2_G &= \lambda^2 \|Xx\|^2_G + (1-\lambda)^2 \|Yx\|^2_G + 2\lambda(1-\lambda)\langle xXGY, x \rangle \\
&= -\lambda(1-\lambda)\|(X-Y)x\|^2_G \\
&\quad + \lambda\|Xx\|^2_G + (1-\lambda)\|Yx\|^2_G \\
&\leq \lambda\|Xx\|^2_G + (1-\lambda)\|Yx\|^2_G. \qquad\qquad\square
\end{aligned}
$$

## B.2 Technical lemmas to prove Theorem 3

**Lemma 6.** *For all $k \geq 0$, the vectors $y_k - x_*$, $x_k - x_*$ and $v_k - x_*$ belong to $\mathbf{Range}(\mathcal{A}^*)$.*

*Proof.* Note that $x_0 = y_0 = x_0$ and in view of (8) we have $x_* \in x_0 + \mathbf{Range}(\mathcal{A}^*)$. So $y_0 - x_* \in \mathbf{Range}(\mathcal{A}^*)$, $v_0 - x_* \in \mathbf{Range}(\mathcal{A}^*)$ and $x_0 - x_* \in \mathbf{Range}(\mathcal{A}^*)$. Assume by induction that $y_k - x_* \in \mathbf{Range}(\mathcal{A}^*)$, $v_k - x_* \in \mathbf{Range}(\mathcal{A}^*)$ and $x_k - x_* \in \mathbf{Range}(\mathcal{A}^*)$. Since $g_k \in \mathbf{Range}(\mathcal{A}^*)$ and $x_{k+1} = y_k - g_k$ we have

$$x_{k+1} - x_* = (y_k - x_*) - g_k \in \mathbf{Range}(\mathcal{A}^*).$$

Moreover,

$$v_{k+1} - x_* = \beta(v_k - x_*) + (1-\beta)(y_k - x*) - \gamma g_k \in \mathbf{Range}(\mathcal{A}^*).$$

Finally

$$y_{k+1} - x_* = \alpha v_{k+1} + (1-\alpha)x_{k+1} - x_* = \alpha(v_{k+1} - x_*) + (1-\alpha)(x_{k+1} - x_*) \in \mathbf{Range}(\mathcal{A}^*).$$

$\qquad\square$

**Lemma 7.**

$$\mathbf{E}\left[ \|Z_k(y_k - x_*)\|^2_{\mathbf{E}[Z]^\dagger} \mid y_k \right] \leq \nu \|y_k - x_*\|^2_{\mathbf{E}[Z]} \tag{28}$$

*Proof.* Since $y_k - x_* \in \mathbf{Range}(\mathcal{A}^*)$ we have that

$$
\begin{aligned}
\mathbf{E}\left[ \|Z_k(y_k - x_*)\|^2_{\mathbf{E}[Z]^\dagger} \mid y_k \right] &= \langle \mathbf{E}\left[ Z_k\mathbf{E}[Z]^\dagger Z_k \right](y_k - x_*), (y_k - x_*) \rangle \\
&\overset{(13)}{\leq} \nu\langle \mathbf{E}[Z](y_k - x_*), (y_k - x_*) \rangle \\
&= \nu\|y_k - x_*\|^2_{\mathbf{E}[Z]}.
\end{aligned}
$$

$\qquad\square$

**Lemma 8.**

$$\|y_k - x_*\|^2_{\mathbf{E}[Z]} = \|y_k - x_*\|^2 - \mathbf{E}\left[ \|x_{k+1} - x_*\|^2 \mid y_k \right] \tag{29}$$

*Proof.*

$$
\begin{aligned}
\mathbf{E}\left[\|x_{k+1} - x_*\|^2 \,|\, y_k\right] &= \mathbf{E}\left[\|(I - Z_k)(y_k - x_*)\|^2 \,|\, y_k\right] \\
&= \langle (I - \mathbf{E}\,[Z])(y_k - x_*), y_k - x_* \rangle \\
&= \|y_k - x_*\|^2 - \|y_k - x_*\|^2_{\mathbf{E}[Z]}.
\end{aligned}
$$

$\square$

### B.3 Proof of Theorem 3

Let $r_k \stackrel{\text{def}}{=} \|v_k - x_*\|^2_{\mathbf{E}[Z]^\dagger}$. It follows that

$$
\begin{aligned}
r_{k+1}^2 &= \|v_{k+1} - x_*\|^2_{\mathbf{E}[Z]^\dagger} \\
&= \|\beta v_k + (1-\beta)y_k - x_* - \gamma Z_k(y_k - x_*)\|^2_{\mathbf{E}[Z]^\dagger} \\
&= \underbrace{\|\beta v_k + (1-\beta)y_k - x_*\|^2_{\mathbf{E}[Z]^\dagger}}_{I} + \gamma^2 \underbrace{\|Z_k(y_k - x_*)\|^2_{\mathbf{E}[Z]^\dagger}}_{II} \\
&\qquad - 2\gamma \underbrace{\langle \beta(v_k - x_*) + (1-\beta)(y_k - x_*), \mathbf{E}\,[Z]^\dagger Z_k(y_k - x_*) \rangle}_{III} \\
&= I + \gamma^2 II - 2\gamma III. \tag{30}
\end{aligned}
$$

The first term can be upper bounded as follows

$$
\begin{aligned}
I &= \|\beta(v_k - x_*) + (1-\beta)(y_k - x_*)\|^2_{\mathbf{E}[Z]^\dagger} \\
&= \beta^2\|v_k - x_*\|^2_{\mathbf{E}[Z]^\dagger} + (1-\beta)^2\|y_k - x_*\|^2_{\mathbf{E}[Z]^\dagger} + 2\beta(1-\beta)\langle v_k - x_*, y_k - x_* \rangle_{\mathbf{E}[Z]^\dagger} \\
&\stackrel{(32)}{=} \beta\|v_k - x_*\|^2_{\mathbf{E}[Z]^\dagger} + (1-\beta)\|y_k - x_*\|^2_{\mathbf{E}[Z]^\dagger} - \beta(1-\beta)\|v_k - y_k\|^2_{\mathbf{E}[Z]^\dagger} \\
&\leq \beta r_k^2 + (1-\beta)\|y_k - x_*\|^2_{\mathbf{E}[Z]^\dagger}, \tag{31}
\end{aligned}
$$

where in the third equality we used a form of the parallelogram identity

$$
2\langle u, v \rangle = \|u\|^2 + \|v\|^2 - \|u - v\|^2, \tag{32}
$$

with $u = v_k - x_*$ and $v = y_k - x_*$.

Taking expectation with to $\mathcal{S}_k$ in the third term in (30) gives

$$
\begin{aligned}
\mathbf{E}\left[III \,|\, y_k, v_k, x_k\right] &= \langle \beta v_k + (1-\beta)y_k - x_*, \mathbf{E}\,[Z]^\dagger \mathbf{E}\,[Z]\,(y_k - x_*) \rangle \\
&= \langle \beta v_k + (1-\beta)y_k - x_*, y_k - x_* \rangle \tag{33} \\
&= \langle \beta \left[\frac{1}{\alpha}y_k - \frac{1-\alpha}{\alpha}x_k\right] + (1-\beta)y_k - x_*, y_k - x_* \rangle \\
&= \langle y_k - x_* + \beta\frac{1-\alpha}{\alpha}(y_k - x_k), y_k - x_* \rangle \\
&= \|y_k - x_*\|^2 + \beta\frac{1-\alpha}{\alpha}\langle y_k - x_k, y_k - x_* \rangle \\
&= \|y_k - x_*\|^2 - \beta\frac{1-\alpha}{2\alpha}\left(\|x_k - x_*\|^2 - \|y_k - x_k\|^2 - \|y_k - x_*\|^2\right) \tag{34}
\end{aligned}
$$

where in the second equality (33) we used that $y_k - x_* \in \mathbf{Range}\,(\mathcal{A}^*) \stackrel{(12)}{=} \mathbf{Range}\,(\mathbf{E}\,[Z])$ together with a defining property of pseudoinverse operators $\mathbf{E}\,[Z]^\dagger \mathbf{E}\,[Z]\,w = w$ for all $w \in \mathbf{Range}\,(\mathbf{E}\,[Z])$. In the last equality (34) we used yet again the identity (32) with $u = y_k - x_k$ and $v = y_k - x_*$.

Plugging (31) and (34) into (30) and taking conditional expectation gives

$$\mathbf{E}\left[r_{k+1}^2 \mid y_k, v_k, x_k\right] = I + \gamma^2 \mathbf{E}\left[II \mid y_k\right] - 2\gamma \mathbf{E}\left[III \mid y_k, v_k, x_k\right]$$

$$\overset{(31)+(34)+(28)}{=} \beta r_k^2 + (1-\beta)\|y_k - x_*\|_{\mathbf{E}[Z]^\dagger}^2 + \gamma^2 \nu \|y_k - x_*\|_{\mathbf{E}[Z]}^2$$

$$+ 2\gamma\left(-\|y_k - x_*\|^2 + \beta\frac{1-\alpha}{2\alpha}\left(\|x_k - x_*\|^2 - \|y_k - x_k\|^2 - \|y_k - x_*\|^2\right)\right)$$

$$\overset{(29)+(14)}{\leq} \beta r_k^2 + \frac{1-\beta}{\mu}\|y_k - x_*\|^2 + \gamma^2 \nu\left(\|y_k - x_*\|^2 - \mathbf{E}\left[\|x_{k+1} - x_*\|^2 \mid y_k\right]\right)$$

$$+ 2\gamma\left(-\|y_k - x_*\|^2 + \beta\frac{1-\alpha}{2\alpha}\left(\|x_k - x_*\|^2 - \|y_k - x_*\|^2\right)\right). \tag{35}$$

Therefore we have that

$$\mathbf{E}\left[r_{k+1}^2 + \gamma^2\nu\|x_{k+1} - x_*\|^2 \mid y_k, v_k, x_k\right] \leq \beta\left(r_k^2 + \underbrace{\gamma\frac{1-\alpha}{\alpha}}_{P_1}\|x_k - x_*\|^2\right)$$

$$+ \left(\underbrace{\frac{1-\beta}{\mu} - 2\gamma + \gamma^2\nu - \beta\gamma\frac{1-\alpha}{\alpha}}_{P_2}\right)\|y_k - x_*\|^2.$$

To establish a recurrence, we need to choose the free parameters $\gamma, \alpha$ and $\beta$ so that $P_1 = \gamma^2\nu$ and $P_2 = 0$. Furthermore we should try to set $\beta$ as small as possible so as to have a fast rate of convergence. Choosing $\beta = 1 - \sqrt{\frac{\mu}{\nu}}, \gamma = \sqrt{\frac{1}{\mu\nu}}, \alpha = \frac{1}{1+\gamma\nu}$ gives $P_2 = 0, \gamma^2\nu = 1/\mu$ and

$$\mathbf{E}\left[r_{k+1}^2 + \frac{1}{\mu}\|x_{k+1} - x_*\|^2 \mid y_k, v_k, x_k\right] \leq \left(1 - \sqrt{\frac{\mu}{\nu}}\right)\left(r_k^2 + \frac{1}{\mu}\|x_k - x_*\|^2\right). \tag{36}$$

Taking expectation and using the tower rules gives the result. $\qquad\square$

## B.4 Changing norm

Given an invertible positive self-adjoint $B \in L(\mathcal{X})$, suppose we want to find the least norm solution of (7) under the norm defined by $\|x\|_B \overset{\text{def}}{=} \sqrt{\langle Bx, x\rangle}$ as the metric in $\mathcal{X}$. That is, we want to solve

$$x^* \overset{\text{def}}{=} \arg\min_{x\in\mathcal{X}} \frac{1}{2}\|x - x_0\|_B^2, \quad \text{subject to} \quad \mathcal{A}x = b. \tag{37}$$

By changing variables $x = B^{-1/2}z$ we have that the above is equivalent to solving

$$z^* \overset{\text{def}}{=} \arg\min_{z\in\mathcal{X}} \frac{1}{2}\|z - z_0\|^2, \quad \text{subject to} \quad \mathcal{A}B^{-1/2}z = b, \tag{38}$$

with $x^* = B^{-1/2}z^*$, and $B^{1/2}$ is the unique symmetric square root of $B$ (see Lemma 18). We can now apply Algorithm 1 to solve (38) where $\mathcal{A}B^{-1/2}$ is the system matrix. Let $x_k$ and $v_k$ be the resulting iterates of applying Algorithm 1. To make explicit this change in the system matrix we define the matrix

$$Z_B \overset{\text{def}}{=} B^{-1/2}\mathcal{A}^*\mathcal{S}_k^*(\mathcal{S}_k\mathcal{A}B^{-1}\mathcal{A}^*\mathcal{S}_k^*)^\dagger\mathcal{S}_k\mathcal{A}B^{-1/2},$$

and the constants

$$\mu_B \overset{\text{def}}{=} \inf_{x\in\mathbf{Range}(B^{-1/2}\mathcal{A}^*)} \frac{\langle\mathbf{E}[Z_B]x, x\rangle}{\langle x, x\rangle} \tag{39}$$

and

$$\nu_B \overset{\text{def}}{=} \sup_{x\in\mathbf{Range}(B^{-1/2}\mathcal{A}^*)} \frac{\langle\mathbf{E}\left[Z_B\mathbf{E}[Z_B]^\dagger Z_B\right]x, x\rangle}{\langle\mathbf{E}[Z_B]x, x\rangle}. \tag{40}$$

Theorem 3 then guarantees that

$$\mathbf{E}\left[\|v_{k+1} - z_*\|^2_{\mathbf{E}[Z_B]^\dagger} + \frac{1}{\mu_B}\|x_{k+1} - z_*\|^2\right] \leq \left(1 - \sqrt{\frac{\mu_B}{\nu_B}}\right)\mathbf{E}\left[\|v_k - z_*\|^2_{\mathbf{E}[Z_B]^\dagger} + \frac{1}{\mu_B}\|x_k - z_*\|^2\right].$$

Reversing our change of variables $\bar{x}_k = B^{-1/2}x_k$ and $\bar{v}_k = B^{-1/2}v_k$ in the above displayed equation gives

$$\mathbf{E}\left[\|\bar{v}_{k+1} - x_*\|^2_{B^{1/2}\mathbf{E}[Z_B]^\dagger B^{1/2}} + \frac{1}{\mu_B}\|\bar{x}_{k+1} - x_*\|^2_B\right]$$

$$\leq \left(1 - \sqrt{\frac{\mu_B}{\nu_B}}\right)\mathbf{E}\left[\|\bar{v}_k - x_*\|^2_{B^{1/2}\mathbf{E}[Z_B]^\dagger B^{1/2}} + \frac{1}{\mu_B}\|\bar{x}_k - x_*\|^2_B\right]. \qquad (41)$$

Thus we recover the same exact from the main theorem in [27], but in a much more general setting.

## C  Proof of Corollary 4

Clearly, $Z = \frac{1}{A_{i,i}}A^{\frac{1}{2}}SS^\top A^{\frac{1}{2}}$, and hence $\mathbf{E}[Z] = \frac{A}{\mathbf{Tr}(A)}$ and $\mu^P = \frac{\lambda_{\min}(A)}{\mathbf{Tr}(A)}$. After simple algebraic manipulations we get

$$\mathbf{E}\left[\mathbf{E}[Z]^{-\frac{1}{2}}Z\mathbf{E}[Z]^{-1}Z\mathbf{E}[Z]^{-\frac{1}{2}}\right] = \mathbf{Tr}(A)^2\,\mathbf{E}\left[\frac{1}{A_{i,i}^2}SS^\top SS^\top\right] = \mathbf{Tr}(A)\,\mathbf{Diag}\left(A_{i,i}^{-1}\right),$$

and therefore $\nu^P = \lambda_{\max}\mathbf{E}\left[\mathbf{E}[Z]^{-\frac{1}{2}}Z\mathbf{E}[Z]^{-1}Z\mathbf{E}[Z]^{-\frac{1}{2}}\right] = \frac{\mathbf{Tr}(A)}{\min_i A_{i,i}}$.

## D  Adding a stepsize $\omega$

In this section we enrich Algorithm 1 with several *additional* parameters and study their effect on convergence of the resulting method.

First, we consider an extension of Algorithm 1 to a variant which uses a *stepsize parameter* $0 < \omega < 2$. That is, instead of performing the update

$$x_{k+1} = y_k - g_k, \qquad (42)$$

we perform the update

$$x_{k+1} = y_k - \omega g_k. \qquad (43)$$

Parameters $\alpha, \beta, \gamma$ are adjusted accordingly. The resulting method enjoys the rate $\mathcal{O}\left(\left(1 - \sqrt{\frac{\nu}{\mu}\omega(2-\omega)}\right)^k\right)$, recovering the rate from Theorem 3 as a special case for $\omega = 1$. The formal statement follows.

**Theorem 9.** *Let $0 < \omega < 2$ be an arbitrary stepsize and define*

$$\eta \overset{def}{=} 2\omega - \omega^2 \geq 0. \qquad (44)$$

*Consider a modification of Algorithm 1 where instead of* (42) *we perform the update* (43). *If we use the parameters*

$$\alpha = \frac{1}{1+\gamma\nu} \qquad\qquad \beta = 1 - \sqrt{\frac{\mu\eta}{\nu}} \qquad\qquad \gamma = \sqrt{\frac{\eta}{\mu\nu}}, \qquad (45)$$

*then the iterates $\{v_k, x_k\}_{k\geq0}$ of Algorithm 1 satisfy*

$$\mathbf{E}\left[\|v_k - x_*\|^2_{\mathbf{E}[Z]^\dagger} + \frac{1}{\mu}\|x_k - x_*\|^2\right] \leq \left(1 - \sqrt{\frac{\mu\eta}{\nu}}\right)^k\mathbf{E}\left[\|v_0 - x_*\|^2_{\mathbf{E}[Z]^\dagger} + \frac{1}{\mu}\|x_0 - x_*\|^2\right].$$

*Proof.* See Appendix F. $\qquad\qquad\qquad\qquad\qquad\qquad\qquad\qquad\qquad\qquad\qquad\qquad\qquad\square$

# E    Allowing for different $\alpha$

In this section we study how the choice of the key parameter $\alpha$ affects the convergence rate.

This parameter determines how much the sequence $y_k = \alpha v_k + (1-\alpha)x_k$ resembles the sequence given by $x_k$ or by $v_k$. For instance, when $\alpha = 0$, $y_k \equiv x_k$, i.e., we recover the steps of the non-accelerated method, and thus one would expect to obtain the same convergence rate as the non-accelerated method. Similar considerations hold in the other extreme, when $\alpha \to 1$. We investigate this hypothesis, and especially discuss how $\beta$ and $\gamma$ must be chosen as a function of $\alpha$ to ensure convergence.

The following statement is a generalization of Theorem 3. For simplicity, we assume that the optional stepsize that was introduced in Theorem 9 is set to one again, $\omega \equiv 1$.

**Theorem 10.** *Let $0 < \alpha < 1$ be fixed. Then the iterates $\{v_k, x_k\}_{k \geq 0}$ of Algorithm 1 with parameters*

$$\beta(s) = \frac{1 + s - s\sqrt{\frac{\nu + 4\mu s - 2\nu s + \nu s^2}{\nu s^2}}}{2s}, \qquad \gamma(s) = \frac{1}{(1 - s\beta(s))\nu}. \qquad (46)$$

*where $\tau \overset{def}{=} \frac{1-\alpha}{\alpha}$ and $s \overset{def}{=} \frac{\tau}{\beta\gamma}$, satisfy*

$$\mathbf{E}\left[\|v_k - x_*\|^2_{\mathbf{E}[Z]^\dagger} + \gamma\tau\|x_k - x_*\|^2\right] \leq \rho^k \mathbf{E}\left[\|v_0 - x_*\|^2_{\mathbf{E}[Z]^\dagger} + \gamma\tau\|x_0 - x_*\|^2\right].$$

*(or put differently):*

$$\mathbf{E}\left[\|v_k - x_*\|^2_{\mathbf{E}[Z]^\dagger} + (1-\alpha)\gamma\|x_k - x_*\|^2\right] \leq \rho^k \mathbf{E}\left[\|v_0 - x_*\|^2_{\mathbf{E}[Z]^\dagger} + (1-\alpha)\gamma\|x_0 - x_*\|^2\right].$$

*where $\rho = \max\{\beta(s), s\beta(s)\} \leq 1$.*

We can now exemplify a few special parameter settings.

**Example 11.** *For $\alpha = 1$, i.e., if $s \to 0$, we get the rate $\rho = 1 - \frac{\mu}{\nu}$ with $\beta = 1 - \frac{\mu}{\nu}$, $\gamma = \frac{1}{\nu}$.*

**Example 12.** *For $\alpha \to 0$, i.e., in the limit $s \to \infty$, we get the rate $\rho = 1 - \frac{\mu}{\nu}$.*

**Example 13.** *The rate $\rho$ is minimized for $s = 1$, i.e., $\beta = 1 - \sqrt{\frac{\nu}{\mu}}$ and $\gamma = \sqrt{\frac{1}{\mu\nu}}$; recovering Theorem 3.*

The best case, in terms of convergence rate for both non-unit stepsize and a variable parameter choice happened to be the default parameter setup. The non-optimal parameter choice was studied in order to have theoretical guarantees for a wider class of parameters, as in practice one might be forced to rely on sub-optimal / inexact parameter choices.

# F    Proof of Theorem 9

The proof follows by slight modifications of the proof of Theorem 3.

First we adapt Lemma 8. As we have $x_{k+1} - x_* = (1 - \omega Z_k)(y_k - x_*)$ the following statement follows by the same arguments as in the proof of Lemma 8.

**Lemma 14** (Lemma 8').

$$\eta\|y_k - x_*\|^2_{\mathbf{E}[Z]} = \|y_k - x_*\|^2 - \mathbf{E}\left[\|x_{k+1} - x_*\|^2 \mid y_k\right] \qquad (47)$$

*Proof.*

$$
\begin{aligned}
\mathbf{E}\left[\|x_{k+1} - x_*\|^2 \mid y_k\right] &= \mathbf{E}\left[\|(I - Z_k)(y_k - x_*)\|^2 \mid y_k\right] \\
&= \mathbf{E}\left[\langle(I - \omega Z_k)(y_k - x_*), (I - \omega Z_k)y_k - x_*\rangle\right] \\
&= \|y_k - x_*\|^2 - \eta\|y_k - x_*\|^2_{\mathbf{E}[Z]}.
\end{aligned}
$$

□

We now follow the same steps as in proof of Theorem 3 in Section B.3. We observe, that the first time Lemma 8 is applied is in equation (35). Using Lemma 14 instead, gives

$$
\mathbf{E}\left[r_{k+1}^2 \mid y_k, v_k, x_k\right] \quad \leq \quad \beta r_k^2 + \frac{1-\beta}{\mu}\|y_k - x_*\|^2 + \frac{\gamma^2 \nu}{\eta}\left(\|y_k - x_*\|^2 - \mathbf{E}\left[\|x_{k+1} - x_*\|^2 \mid y_k\right]\right)
$$

$$
+ 2\gamma\left(-\|y_k - x_*\|^2 + \beta\frac{1-\alpha}{2\alpha}\left(\|x_k - x_*\|^2 - \|y_k - x_*\|^2\right)\right). \qquad (48)
$$

Therefore we have that

$$
\mathbf{E}\left[r_{k+1}^2 + \gamma^2 \nu\|x_{k+1} - x_*\|^2 \mid y_k, v_k, x_k\right] \quad \leq \quad \beta\left(r_k^2 + \underbrace{\gamma\frac{1-\alpha}{\alpha}}_{P_1'}\|x_k - x_*\|^2\right)
$$

$$
+ \left(\underbrace{\frac{1-\beta}{\mu} - 2\gamma + \frac{\gamma^2 \nu}{\eta} - \beta\gamma\frac{1-\alpha}{\alpha}}_{P_2'}\right)\|y_k - x_*\|^2.
$$

Noting that $\frac{1-\alpha}{\alpha} = \gamma\nu$ and $\frac{\gamma^2\nu}{\eta} = \frac{\gamma(1-\alpha)}{\eta\alpha} = \frac{1}{\mu}$, we observe $P_2' = 0$ and deduce the statement of Theorem 9.

## G  Proof of Theorem 10

It suffices to study equation (35). We observe that for convergence the big bracket, $P_2$, should be negative,

$$
(1-\beta)\frac{1}{\mu} + \gamma^2\nu - 2\gamma - \gamma\beta\frac{1-\alpha}{\alpha} \leq 0 \qquad (49)
$$

The convergence rate is then

$$
\rho \stackrel{\text{def}}{=} \max\left\{\beta, \frac{(1-\alpha)\beta}{\alpha\gamma\nu}\right\}. \qquad (50)
$$

or in the notation of Theorem 10, $\rho = \max\{\beta, s\beta\}$.

This means, that in order to obtain the best convergence rate, we should therefore choose parameters $\beta$ and $\gamma$ such that $\beta$ is as small as possible. This observation is true regardless of the value of $s$ (which itself depends on $\gamma$).

With the notation $\tau = s\gamma\beta$, we reformulate (49) to obtain

$$
\frac{1}{\mu} + \gamma^2\nu - 2\gamma \leq \beta\left(\frac{1}{\mu} + s\gamma^2\nu\right) \qquad (51)
$$

Thus we see, that $\beta$ cannot be chosen smaller than

$$
\beta^\star(s, \gamma) = \frac{1 + \mu\gamma^2\nu - 2\mu\gamma}{1 + s\mu\gamma^2\nu} \qquad (52)
$$

Minimizing this expression in $\gamma$ gives

$$
\beta^\star(s) = \frac{1 + s - s\sqrt{\frac{\nu + 4\mu s - 2\nu s + \nu s^2}{\nu s^2}}}{2s} \qquad (53)
$$

with $\gamma^\star(s) = \frac{1}{(1 - s\beta^\star(s))\nu}$.

We further observe that this parameter setting indeed guarantees convergence, i.e. $\rho \leq 1$. From (53) we observe ($\nu > 0$, $s \geq 0$, $\mu \geq 0$):

$$
\beta^\star(s) \leq \frac{1 + s - \sqrt{\frac{\nu - 2\nu s + \nu s^2}{\nu}}}{2s} = \frac{1 + s - (s-1)}{2s} = \frac{1}{s} \qquad (54)
$$

Hence $s\beta^\star(s) \leq 1$. On the other hand, $(1-s) \leq \sqrt{(1-s)^2 + \frac{4\mu s}{\nu}}$ and hence $(1+s) - \sqrt{(1-s)^2 + \frac{4\mu s}{\nu}} \leq 2s$, which shows $\beta^\star(s) \leq 1$.

# H    Proofs and Further Comments on Section 4

## H.1    Proof of Theorem 5

We perform a change of coordinates since it is easier to work with the standard Frobenius norm as opposed to the weighted Frobenius norm. Let $\hat{X} = A^{1/2}XA^{1/2}$ so that (18) and (20) become

$$\hat{X}_* \stackrel{\text{def}}{=} I = \arg\min\|\hat{X}\|_F^2 \quad \text{subject to} \quad \hat{X} = I, \quad \hat{X} = \hat{X}^\top, \tag{55}$$

and

$$\hat{X}_{k+1} = P + (I - P)\,\hat{X}_k\,(I - P)\,, \tag{56}$$

respectively, where $P = A^{1/2}S(S^\top AS)^{-1}S^\top A^{1/2}$. The linear operator that encodes the constraint in (4.2) is given by $\hat{\mathcal{A}}(X) = \left(X,\ X - X^\top\right)$ the adjoint of which is given by $\hat{\mathcal{A}}^*(Y_1, Y_2) = Y_1 + Y_2 - Y_2^\top$. Since $\hat{\mathcal{A}}^*$ is clearly surjective, it follows that $\mathbf{Range}\left(\hat{\mathcal{A}}^*\right) = \mathbb{R}^{n\times n}$.

Subtracting the identity matrix from both sides of (56) and using that $P$ is a projection matrix, we have that

$$\hat{X}_{k+1} - I = (I - P)(\hat{X}_k - I)(I - P)\,. \tag{57}$$

To determine the $Z$ operator (9), from (11) and (57) we know that

$$(I - P)(\hat{X}_k - I)(I - P) = (I - Z)(\hat{X}_k - I).$$

Thus for every matrix $X \in \mathbb{R}^{n\times n}$ we have that

$$Z(X) = X - (I - P)X(I - P) = XP + PX(I - P). \tag{58}$$

Denote column-wise vectorization of $X$ as $x$: $x \stackrel{\text{def}}{=} \mathbf{Vec}\,(X)$. To calculate a useful lower bound on $\mu$, note that

$$
\begin{aligned}
\mathbf{Tr}\left(X^\top Z(X)\right) &= \mathbf{Tr}\left(X^\top XP\right) + \mathbf{Tr}\left(X^\top PX(I - P)\right)\\
&= x^\top \mathbf{Vec}\,(XP) + x^\top \mathbf{Vec}\,(PX(I - P))\\
&= x^\top(P \otimes I)x + x^\top((I - P) \otimes P)x\\
&\stackrel{(23)}{=} x^\top \mathbf{Z}x,
\end{aligned}
\tag{59}
$$

where we used that $\mathbf{Tr}\left(A^\top B\right) = \mathbf{Vec}\,(A)^\top \mathbf{Vec}\,(B)$ and $\mathbf{Vec}\,(AXB) = (B^\top \otimes A)\mathbf{Vec}\,(x)$ holds for any $A, B, X$.

Consequently, $\mu$ is equal to

$$\mu \stackrel{(13)}{=} \inf_{X \in \mathbb{R}^{n\times n}} \frac{\langle \mathbf{E}\,[Z]\,X, X\rangle_F}{\|X\|_F^2} \stackrel{(59)}{=} \inf_{x \in \mathbb{R}^{n^2 \times n^2}} \frac{x^\top \mathbf{E}\,[\mathbf{Z}]\,x}{x^\top x} = \lambda_{\min}(\mathbf{E}\,[\mathbf{Z}]).$$

Notice that we have $2\lambda_{\min}(\mathbf{E}\,[P]) \geq \lambda_{\min}(\mathbf{E}\,[\mathbf{Z}]) \geq \lambda_{\min}(\mathbf{E}\,[P])$ since $(P \otimes I) + (I \otimes P) \geq \mathbf{Z} \geq (P \otimes I)$.

In light of Algorithm 1, the iterates of the accelerated version of (56) are given by

$$
\begin{aligned}
\hat{Y}_k &= \alpha\hat{V}_k + (1 - \alpha)\hat{X}_k\\
\hat{G}_k &= Z_k(\hat{Y}_k - I)\\
\hat{X}_{k+1} &= \hat{Y}_k - \hat{G}_k\\
\hat{V}_{k+1} &= \beta\hat{V}_k + (1 - \beta)\hat{Y}_k - \gamma\hat{G}_k
\end{aligned}
\tag{60}
$$

where $\hat{Y}_k, \hat{V}_k, \hat{G} \in \mathbb{R}^{n\times n}$. From Theorem 3 we have that $\hat{V}_k$ and $\hat{X}_k$ converge to the identity matrix according to

$$\mathbf{E}\left[\|\hat{V}_{k+1} - I\|_{\mathbf{E}[Z]^\dagger}^2 + \frac{1}{\mu}\|\hat{X}_{k+1} - I\|_F^2\right] \leq \left(1 - \sqrt{\frac{\mu}{\nu}}\right)\mathbf{E}\left[\|\hat{V}_k - I\|_{\mathbf{E}[Z]^\dagger}^2 + \frac{1}{\mu}\|\hat{X}_k - I\|_F^2\right],$$
$$\tag{61}$$

where $\|X\|_{\mathbf{E}[Z]^\dagger}^2 = \langle \mathbf{E}\,[Z]^\dagger X, X\rangle_F$. Changing coordinates back to $\hat{X}_k = A^{1/2}X_k A^{1/2}$ and defining $Y_k \stackrel{\text{def}}{=} A^{-1/2}\hat{Y}_k A^{-1/2}$, $V_k \stackrel{\text{def}}{=} A^{-1/2}\hat{V}_k A^{-1/2}$ and $G_k \stackrel{\text{def}}{=} A^{-1/2}\hat{G}_k A^{-1/2}$, we have that (61) gives (21). Furthermore, using the same coordinate change applied to the iterates (60) gives Algorithm 2.

## H.2 Matrix inversion as linear system

Denote $x = \mathbf{Vec}\,(X)$, i.e. $x$ is $n^2$ dimensional vector such that $X_{(n(i-1)+1):ni} = X_{:,i}$. Similarly, denote $e = \mathbf{Vec}\,(I)$. System (6) can be thus rewritten as

$$(I \otimes A)x = e. \tag{62}$$

Notice that all linear sketches of the original system $AX = I$ can be written as

$$S_0^\top (I \otimes A)x = S_0^\top e \tag{63}$$

for a suitable $n^2 \times n^2$ matrix $S_0$, therefore the setting is fairly general.

### H.2.1 Alternative proof of Theorem 5

Let us now, for a purpose of this proof, consider sketch matrix $S_0$ to capture only sketching the original matrix system $AX = I$ by left multiplying by $S$, i.e. $S_0 = (I \otimes S)$, as those are the considered sketches in the setting of Section 4.

As we have

$$\mathbf{Tr}\left(BX^\top BX\right) = \mathbf{Vec}\,(BXB)^\top x = x^\top (B \otimes B)x,$$

weighted Frobenius norm of matrices is equivalent to a special weighted euclidean norm of vectors. Define also $C$ to be a matrix such that $Cx = 0$ if and only if $X = X^\top$. Therefore, (4.2) is equivalent to

$$x_{k+1} = \arg\min \|x - x_k\|_{A \otimes A}^2 \quad \text{subject to} \quad (I \otimes S^\top)(I \otimes A)x = (I \otimes S^\top)e, \quad Cx = 0, \tag{64}$$

which is a sketch-and-project method applied on the linear system, with update as per (20):

$$x^{k+1} = x^k - (H \otimes I)((I \otimes A)x - e) - (I \otimes H)((I \otimes A)x - e) + (HA \otimes H)((I \otimes A)x - e)$$

for $H \stackrel{\text{def}}{=} S\left(S^\top A S\right)^{-1} S^\top$. Using substitution $\hat{x} = (A^{\frac{1}{2}} \otimes A^{\frac{1}{2}})x$; $\hat{S} = A^{\frac{1}{2}}S$ and comparing to (11), we get

$$Z = I \otimes I - (I - P) \otimes (I - P)$$

for $P$ as defined inside the statement of Theorem 5. Therefore, we have all necessary information to apply the results from [27], recovering Theorem 5.

# I  Linear Operators in Euclidean Spaces

Here we provide some technical lemmas and results for linear operators in Euclidean space, that we used in the main body of the paper. Most of these results can be found in standard textbooks of analysis, such as [25]. We give them here for completion.

Let $\mathcal{X}, \mathcal{Y}, \mathcal{Z}$ be Euclidean spaces, equipped with inner products. Formally, we should use a notation that distinguishes the inner product in each space. But instead we use $\langle \cdot, \cdot \rangle$ to denote the inner product on all spaces, as it will be easy to determine from which space the elements are in. That is, for $x_1, x_2 \in \mathcal{X}$, we denote by $\langle x_1, x_2 \rangle$ the inner product between $x_1$ and $x_2$ in $\mathcal{X}$.

Let

$$\|T\| \stackrel{\text{def}}{=} \sup_{\|x\| \leq 1} \|Tx\|,$$

denote the operator norm of $T$. Let $0 \in L(\mathcal{X}, \mathcal{Y})$ denote the zero operator and $I \in L(\mathcal{X}, \mathcal{Y})$ the identity map.

**The adjoint.** Let $T^* \in L(\mathcal{Y}, \mathcal{X})$ denote the unique operator that satisfies

$$\langle Tx, y \rangle = \langle x, T^*y \rangle,$$

for all $x \in \mathcal{X}$ and $y \in \mathcal{Y}$. We say that $T^*$ is the *adjoint* of $T$. We say $T$ is *self-adjoint* if $T = T^*$. Since for all $x \in \mathcal{X}$ and $s \in \mathcal{S}$,

$$\langle x, (ST)^*s \rangle = \langle STx, s \rangle_\mathcal{S} = \langle Tx, S^*s \rangle_\mathcal{Y} = \langle x, T^*S^*s \rangle,$$

we have

$$(ST)^* = T^*S^*.$$

**Lemma 15.** *For $T \in L(\mathcal{X}, \mathcal{Y})$ we have that* $\mathbf{Range}\,(T^*)^{\perp} = \mathbf{Null}\,(T)$. *Thus*

$$
\begin{aligned}
\mathcal{X} &= \mathbf{Range}\,(T^*) \oplus \mathbf{Null}\,(T) & (65) \\
\mathcal{Y} &= \mathbf{Range}\,(T) \oplus \mathbf{Null}\,(T^*) & (66)
\end{aligned}
$$

*Proof.* See 3.2.6 in [25]. □

## I.1 Positive Operators

We say that $G \in L(\mathcal{X})$ is positive if it is self-adjoint and if $\langle x, Gx \rangle \geq 0$ for all $x \in \mathcal{X}$. Let $(e_j)_{j=1}^{\infty} \in \mathcal{X}$ be an orthonormal basis. The trace of $G$ is defined as

$$
\mathbf{Tr}\,(G) \stackrel{\text{def}}{=} \sum_{j=1}^{\infty} \langle Ge_j, e_j \rangle. \tag{67}
$$

The definition of trace is independent of the choice of basis due to the following lemma.

**Lemma 16.** *If $U$ is unitary and $G \geq 0$ then* $\mathbf{Tr}\,(UGU^*) = \mathbf{Tr}\,(G)$.

*Proof.* See 3.4.3 and 3.4.4 in [25]. □

**Lemma 17.** *If $P \in L(\mathcal{X})$ is a projection matrix then* $\mathbf{Tr}\,(P) = \dim(\mathbf{Range}\,(P)) = \mathbf{Rank}\,(P)$.

*Proof.* Let $d = \dim(\mathbf{Range}\,(P))$ which is possibly infinite. Given that $P$ is a projection we have that $\mathbf{Range}\,(P)$ is a closed subspace and thus there exists orthonormal basis $(e_j)_{j=1}^{d}$ of $\mathbf{Range}\,(P)$. Consequently, $\mathbf{Tr}\,(P) \stackrel{(67)}{=} \sum_{j=1}^{d} 1 = d = \dim(\mathbf{Range}\,(P))$. □

A *square root* of an operator $G \in L(\mathcal{X})$ is an operator $R \in L(\mathcal{X})$ such that $R^2 = G$.

**Lemma 18.** *If $G : \mathcal{X} \to \mathcal{X}$ is positive, then there exists a unique positive square root of $G$ which we denote by $G^{1/2}$.*

*Proof.* See 3.2.11 in [25]. □

**Lemma 19.** *For any $T \in L(\mathcal{X}, \mathcal{Y})$ and any $G \in L(\mathcal{Y}, \mathcal{Y})$ that is positive and injective,*

$$
\mathbf{Null}\,(T) = \mathbf{Null}\,(T^*GT), \tag{68}
$$

*and*

$$
\overline{\mathbf{Range}\,(T^*)} = \overline{\mathbf{Range}\,(T^*GT)}. \tag{69}
$$

*Proof.* The inclusion $\mathbf{Null}\,(T) \subset \mathbf{Null}\,(T^*GT)$ is immediate. For the opposite inclusion, let $x \in \mathbf{Null}\,(T^*GT)$. Since $G$ is positive we have by Lemma 18 that there exists a square root with $G^{1/2}G^{1/2} = G$. Therefore, $\langle x, T^*GTx \rangle = \langle G^{1/2}Tx, G^{1/2}Tx \rangle = 0$, which implies that $G^{1/2}Tx = 0$. Since $G$ is injective, it follows that $G^{1/2}$ is injective and thus $x \in \mathbf{Null}\,(T)$. Finally (69) follows by taking the orthogonal complements of (68) and observing Lemma 15. □

As an immediate consequence of (68) and (69) we have the following lemma.

**Corollary 20.** *For $G : \mathcal{X} \to \mathcal{X}$ positive we have that*

$$
\begin{aligned}
\mathbf{Null}\,\left(G^{1/2}\right) &= \mathbf{Null}\,(G) & (70) \\
\overline{\mathbf{Range}\,\left(G^{1/2}\right)} &= \overline{\mathbf{Range}\,(G)} & (71)
\end{aligned}
$$

### I.2 Pseudoinverse

For a bounded linear operator $T$ define the pseudoinverse of $T$ as follows.

**Definition 21.** *Let $T \in L(\mathcal{X}, \mathcal{Y})$ such that $\mathbf{Range}\,(T)$ is closed. $T^\dagger : \mathcal{Y} \to \mathcal{X}$ is said to be the pseudoinverse if*

i) $T^\dagger T x = x$ *for all* $x \in \mathbf{Range}\,(T^*)$.

ii) $T^\dagger x = 0$ *for all* $x \in \mathbf{Null}\,(T^*)$.

iii) *If* $x \in \mathbf{Null}\,(T)$ *and* $y \in \mathbf{Range}\,(T^*)$ *then* $T^\dagger (x + y) = T^\dagger x + T^\dagger y$.

It follows directly from the definition (see [9] for details) that $T^\dagger$ is a unique bounded linear operator. The following properties of pseudoinverse will be important.

**Lemma 22** (Properties of pseudoinverse). *Let $T \in L(\mathcal{X}, \mathcal{Y})$ such that $\mathbf{Range}\,(T)$ is closed. It follows that*

i) $TT^\dagger T = T$

ii) $\mathbf{Range}\,\left(T^\dagger\right) = \mathbf{Range}\,(T^*)$ *and* $\mathbf{Null}\,\left(T^\dagger\right) = \mathbf{Null}\,(T^*)$

iii) $(T^*)^\dagger = (T^\dagger)^*$

iv) *If $T$ is self-adjoint and positive then $T^\dagger$ is self-adjoint and positive.*

v) $T^\dagger TT^* = T^*$, *that is, $T^\dagger T$ projects orthogonally onto $\mathbf{Range}\,(T^*)$ and along $\mathbf{Null}\,(T)$.*

vi) *Consider the linear system $Tx = d$ where $d \in \mathbf{Range}\,(T)$. It follows that*

$$T^\dagger d = \arg\min_{x \in \mathcal{X}} \tfrac{1}{2}\|x\|^2 \quad \text{subject to} \quad Tx = d. \tag{72}$$

vii) $T^\dagger = T^*(TT^*)^\dagger$

*Proof.* The proof of items *i, ii, iii, iv, v* can be found in [9]. The proof of item *vi* is alternative characterization of the pseudoinverse and it can be established by using that $d \in \mathbf{Range}\,(T)$ together with item *i* thus $TT^\dagger d = d$. The proof then follows by using the orthogonal decomposition $\mathbf{Range}\,(T^*) \oplus \mathbf{Null}\,(T)$ to show that $T^\dagger d$ is indeed the minimum of (72). Finally item (vii) is a direct consequence of the previous items. $\qquad\square$