[Reviews · NeurIPS 2018]

Reviewer 1



This paper presents an accelerated version of the sketch-and-project algorithm, an accelerated algorithm for matrix inversion and accelerated variants of deterministic and stochastic quasi-Newton updates. I strongly believe that this line of research is of interest for the ML and optimization communities, and that the algorithms and theoretical results presented in this paper are significant and novel. Moreover, the numerical results presented in the paper clearly illustrate the effectiveness of the approaches presented in the paper. For this reason, I strongly recommend this paper for publication. Below I list my minor concerns with the paper. - The paper is clearly written and easy to read, however, I believe that in several places there are grammatical errors or minor typos. These are easy to fix. A few examples are: — Line 25: liner system in each iteration -> linear system at each iteration — Line 27: in particular the BFGS -> in particular BFGS — Line 116: which in term -> which in turn — Line 122-123: Sentence starting with “This is a very general problem…” has a grammatical issue. — Line 142: in the special case -> for the special case — Line 247: inverting a matrix symmetric positive matrix A -> inverting a symmetric positive definite matrix A — Line 263-264: Sentence starting with “We also observe from…” is not clear. — Line 266: as fast as nonaccelerated algorithm -> as fast as the nonaccelerated algorithm - I strongly believe that a Final Remarks and Future Work (Extensions) section is missing from the paper. This section would help the reader summarize the contributions of the paper and understand the significance and possible extensions of this work. - Other Minor Issues: — The authors should explain how alpha, beta and gamma are chosen in the Algorithms, and why these choices were made. — Line 207: Issue with reference to (4.2)? — Line 230: Issue with equation reference? — Reference [32] and [33] is the same — Reference section lacks consistency (names of authors) - I believe that there are some missing references in the paper. — stochastic quasi-Newton methods: 1) A self-correcting variable-metric algorithm for stochastic optimization (http://proceedings.mlr.press/v48/curtis16.pdf), 2) A multi-batch l-bfgs method for machine learning (http://papers.nips.cc/paper/6145-a-multi-batch-l-bfgs-method-for-machine-learning.pdf) — subsampled Newton methods: 1) Sub-sampled Newton methods I: globally convergent algorithms (https://arxiv.org/pdf/1601.04737.pdf), 2) Newton-type methods for non-convex optimization under inexact hessian information (https://arxiv.org/pdf/1708.07164.pdf), 3) On the use of stochastic hessian information in optimization methods for machine learning (https://pdfs.semanticscholar.org/ba21/5739785c21704dc2a6c8fa235c52cab0f45d.pdf), (4) Convergence rates of sub-sampled Newton methods (https://arxiv.org/abs/1508.02810)

Reviewer 2



This paper proposed an accelerated randomized algorithm for solving linear systems based on sketching and projection. Then they extended it for matrix inversion problem and also presented an accelerated BFGS optimization algorithm using the same heuristics. 1. This paper claims this is the *first* accelerated randomized algorithm for solving linear system. I am not so sure what is the right baseline rate that this paper tries to accelerate. Usually the non-accelerated is O(\kappa \log(1/\eps)) and the accelerated rate is O(\sqrt\kappa \log(1/\eps)), \kappa is the condition number of A. To my best knowledge, the first accelerated stochastic algorithm that can achieve the latter rate under certain condition is from this paper: http://proceedings.mlr.press/v84/xu18a/xu18a-supp.pdf where they analyzed the general stochastic three-term recurrence. After reading the paper, the paper is accelerating the the specific Sketch-and-Project algorithm which has a rate O(1/\mu\log 1/eps). It is not clear to me how the accelerated rate O(\sqrt(\nu/\mu) \log 1/\eps) from this paper compares with the typical accelerated rate O(\sqrt\kappa \log 1/\eps). To be precise, the paper should not claim this is the first accelerated randomized algorithm, unless it makes it very clear which specific rate/algorithm they try to accelerate. From a more practical point, what are the scenarios where these algorithms can outperform CG or MINRES? 2. In the experiments, it is not clear that what the legends/axis are. 3. I suggest to include CG in the experiments for comparison. 4. Algorithm 3 is very interesting. If the paper could do some analysis on the momentum scheme for BFGS, that is a really big contribution. Unfortunately there is none. Overall, I suggest that the paper should better position their paper, be more descriptive about the experiment and do some analysis on Algorithm 3. Then that will be a very strong paper.

Reviewer 3



This paper first analyzes the Accelerated Sketch-and-Project of ref. [24] in more general cases and then applies that method for Matrix Inversion problem. They show the accelerated convergence rate for the inversion problem. Finally, they propose that the acceleration tweak could be incorporated in the BFGS algorithm. C 1- This is an incremental contribution over the original Accelerated Sketch-and-Project and its application over matrix inversion problem. 2- It is not clear why there is the section 4.2. It needs more explanation. 3- It mentioned in the paper that the assumption(12) holds for the inversion problem. Is it independent of the sketching strategy? How so?